# Understanding Visual Feature Reliance through the Lens of Complexity

**Thomas Fel** [*†]
Google DeepMind
Brown University

**Louis Béthune** [*‡]
Université de Toulouse

**Andrew Kyle Lampinen**
Google DeepMind

**Thomas Serre**
Brown University

**Katherine Hermann**
Google DeepMind

## Abstract

Recent studies suggest that deep learning models' inductive bias towards favoring simpler features may be one of the sources of shortcut learning. Yet, there has been limited focus on understanding the complexity of the myriad features that models learn. In this work, we introduce a new metric for quantifying feature complexity, based on $\mathcal{V}$-information and capturing whether a feature requires complex computational transformations to be extracted. Using this $\mathcal{V}$-information metric, we analyze the complexities of 10,000 features—represented as directions in the penultimate layer—that were extracted from a standard ImageNet-trained vision model. Our study addresses four key questions: First, we ask ***what*** features look like as a function of complexity and find a spectrum of simple-to-complex features present within the model. Second, we ask ***when*** features are learned during training. We find that simpler features dominate early in training, and more complex features emerge gradually. Third, we investigate ***where*** within the network simple and complex features "flow", and find that simpler features tend to bypass the visual hierarchy via residual connections. Fourth, we explore the connection between features' complexity and their importance in driving the network's decision. We find that complex features tend to be less important. Surprisingly, important features become accessible at earlier layers during training, like a "sedimentation process," allowing the model to build upon these foundational elements.

> *"It is necessary to have on hand a method of measuring the complexity of calculating devices which in turn can be done if one has a theory of the complexity of functions, some partial results on this problem have been obtained by Shannon."*

Darmouth Workshop proposal [70]

Measuring complexity is one of the core problems described by Shannon & McCarty in the famous 1956 proposal of the Dartmouth workshop. This problem—and the question, *"How can a set of (hypothetical) neurons be arranged to form concepts?"*[70]—encapsulate what we investigate: how do neural networks form features and concepts [53], and how can their complexity be quantified?

Recent studies [97, 48] reveal that models often favor simpler features, which may contribute to shortcut learning [6, 71, 36, 100]. For example, CNNs privilege texture over object shape [10, 37, 46] and single diagnostic pixels over semantic content [69]. Moreover, models tend to prefer input features that are linearly rather than nonlinearly related to task labels [47, 97]. However, there

---

[*] These authors contributed equally to this work
[†] Work done as Student Researcher at Google DeepMind
[‡] Work completed prior to joining Apple

38th Conference on Neural Information Processing Systems (NeurIPS 2024).

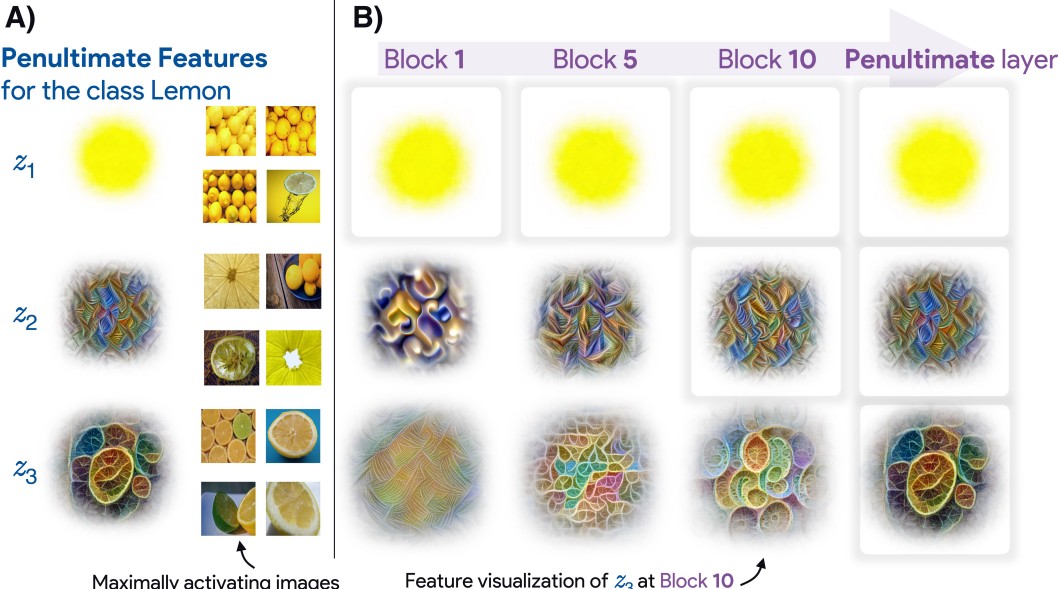

Figure 1: **A) Simple vs. Complex Features.** Shown is an example of three features extracted using an overcomplete dictionary on the penultimate layer of a ResNet50 trained on ImageNet. Although all three features can be extracted from the final layer of a ResNet50, some features, such as $z_1$, seem to respond to color, which can be linearly extractable directly from the input. In contrast, $z_2, z_3$ visualization appear more "Complex", responding to more diverse stimuli. In this work, we seek to study the complexity of features. We start by introducing a computationally inspired complexity metric. Using this metric, we inspect both simple and complex features of a ResNet50. **B) Feature Evolution Across Layers.** Each row illustrates how a feature from the penultimate layer ($z_1, z_2, z_3$) evolves as we decode it using linear probing at the outputs of blocks 1, 5, and 10 of the ResNet50. Simpler features, like color, are decodable throughout the network. The feature in the middle shows similar visualization at block 10 and the penultimate layer, whereas the most complex feature is only decodable at the end. Our complexity metric, based on $\mathcal{V}$-information [115], measures how easily a model extracts a feature across its layers.

has not been a comprehensive quantitative framework for assessing the complexities of features learned by large-scale, natural image-trained vision models used in practice. Leveraging recent observations and advances in feature (also called "concept") extraction from the area of Explainable AI [83, 8, 25, 29, 32, 77, 19, 32, 42], we extract a large set of features from an ImageNet-trained model [45], and analyze their complexity.

Our contributions are as follows:

- We build upon $\mathcal{V}$-information [115]—which measures the mutual information between two variables, considering computational constraints—to introduce a measure of feature complexity. We use this measure to quantify the complexity of over 10,000 features in an ImageNet model [45] at each epoch of training.
- We visualize the differences between simple and complex features on a spectrum to understand which features are readily available to our model and which ones require more computation and transformation to retrieve.
- We investigate **where** sensitivity to simple versus complex features emerges during a forward pass through the model. Our findings suggest that residual connections "teleport" simple features, computed in early layers, to the final layer. The main branch naturally facilitates the layer-wise construction of more complex features.
- We examine feature learning dynamics, revealing **when** different concepts emerge over the course of training, and find that complex concepts tend to appear later than simpler ones.
- We explore the link between complexity and importance in driving the model's decisions. We find a preference for simpler features over more complex ones. This simplicity bias emerges during training, and, surprisingly, the model simplifies its most important features over time.

That is, during training, important features become accessible at an earlier layer (via a shorter computational graph).

# 1  Related Work

**Feature analysis.** Large vision models learn a diversity of features [77, 84] to support performance on the training task and can exhibit preferences for certain features over others, for example textures over shapes [10, 37, 46]. These preferences can be related to their use of shortcuts [36, 100] which compromise generalization capabilities [74, 73]. Hermann et al. [48] suggest that a full account of a model's feature preferences should consider both the predictivity and *availability* of features, and identify image properties that induce a shortcut bias. Relatedly, work shows that models often prefer features that are computationally simpler to extract—a "simplicity bias" [107, 87, 91, 7, 97, 47].

**Explainability.** Attribution methods [99, 117, 9, 33, 88, 80, 41, 102, 106] seek to attribute model predictions to specific input parts and to visualize the most important area of an image for a given prediction. In response to the many limitations of these methods [2, 38, 101, 27, 54, 79], Feature Visualization [78, 84, 44] methods have sought to allow for the generation of images that maximize certain structures in the model – e.g., a single neuron, entire channel, or direction, providing a clearer view features learned early [82, 96], as well as circuits present in the models [83, 62]. Recently, work has scaled these methods to deeper models [31]. Another approach, complementary to feature visualization, is automated concept extraction [39, 119, 30, 32, 1, 112], which identifies a wide range of concepts – directions in activations space – learned by models, inspired by recent works that suggest that the number of learned features often exceeds the neuron count [29]. This move towards over-complete dictionary learning for more comprehensive feature analysis represents a critical advancement.

**Complexity.** On a theoretical level, the complexity of functions in deep learning has long been a subject of interest, with traditional frameworks like VC-dimension falling short of adequacy with current results. In particular, deep learning models often have the capacity to memorize the entire dataset, yet still generalize [118]; the reason is often suggested to be a positive benefit of simplicity bias [3, 51, 110]. Measures of the complexity of neural network functions are hard to make tractable [93]. Recent work has proposed various methods to evaluate this complexity. For instance, [17] proposed a score of non-linearity propagation, while [52] introduced a measure of local complexity based on spline partitioning. Additionally, [111] demonstrated that models tend to learn functions with low sensitivity to random changes in the input. The role of optimizers in complexity has also been explored. It has been shown that different optimizers impact the features learned by models; for example, [105] found that sharpness-aware minimization (SAM) [34] learns more diverse features, both simple and hard, whereas stochastic gradient descent (SGD) models tend to rely on simpler features. Furthermore, [24] utilized category theory to propose a metric based on redundancy, which consist in merging neurons until a distance gap is too large, with this distance gap acting as a hyperparameter. Concurrent work by Lampinen et al. [58] studies representations induced by input features of different complexities when datasets are carefully controlled and manipulated. Finally, Okawa et al. [81], Park et al. [85] investigated the development of concepts during the training process on toy datasets and revealed that the sequence in which they appear, related to their complexity, can be attributed to the multiplicative emergence of compositional skills.

Concerning algorithmic complexity, Kolmogorov complexity [103, 56, 21], later expanded by Levin [63] to include a computational time component, offers a measure for evaluating the shortest programs capable of generating specific outputs on a Turing machine [22, 43]. This notion of complexity is at the roots of Solomonoff induction [104], which is often understood as the formal expression of Occam's razor and has received some attention in deep learning community [94, 95, 16]. Further developing these concepts, $\mathcal{V}$-information [115] introduces computational constraints on mutual information measures, extending Shannon's legacy. This methodology enables the assessment of a feature's availability or the simplicity with which it can be decoded from a data source. We will formally introduce this concept in Section 2.

---

We observe in Appendix E that our complexity measure is also correlated with this category theory based complexity metric.

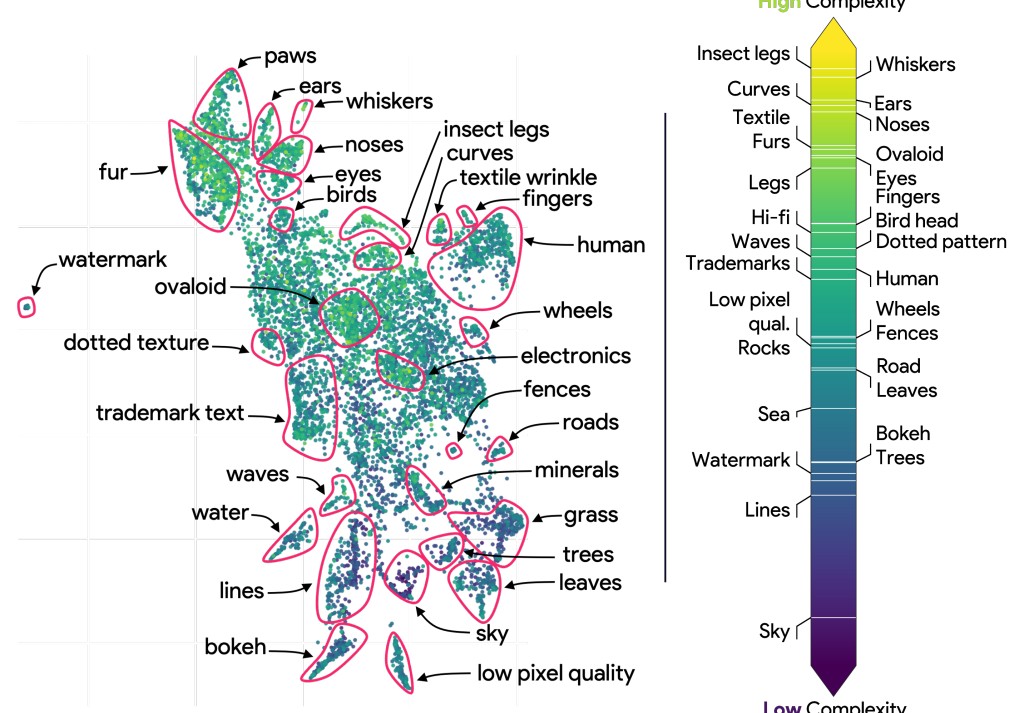

Figure 2: **Qualitative Analysis of "Meta-feature" (cluster of features) Complexity. (Left)** A 2D UMAP projection displaying the 10,000 extracted features. The features are organized into 150 clusters using K-means clustering applied to the feature dictionary $\mathbf{D}^\star$. 30 clusters were selected for analysis of features at different complexity levels. **(Right)** For each Meta-feature cluster, we compute the average complexity score. This allows us to classify the features based on their complexity according to the model. Notably, simple features are often akin to color detectors (e.g., *grass*, *sky*) and detectors for *low-frequency patterns* (e.g., *bokeh* detector) or *lines*. In contrast, complex features encompass parts or structured objects, as well as features resembling shapes (such as *ears* or *curve detectors*). Visualizations of individual Meta-features are presented in Appendix B.

## 2   Method

Before we measure feature complexity, we define what is meant by features, explain how they are extracted, and then introduce the complexity metric.

**Model Setup.** We study feature complexity within an ImageNet-trained ResNet50 [45]. We train the model for 90 epochs with an initial learning rate of 0.7, adjusted down by a factor of 10 at epochs 30, 60, and 80, achieving a 78.9% accuracy on the ImageNet validation set, which is on par with reported accuracy in similar studies [45, 114]. Focusing on one model reduces architectural variables, creating a controlled environment to analyze feature complexities and provide insights for broader model hypotheses.

**Feature Extraction.** We operate within a classical supervised machine learning setting on $(\Omega, \mathcal{F}, \mathbb{P})$ – the underlying probability space – where $\Omega$ is the sample space, $\mathcal{F}$ is a $\sigma$-algebra on $\Omega$, and $\mathbb{P}$ is a probability measure on $\mathcal{F}$. The input space is denoted $\mathcal{X} \subseteq \mathbb{R}^d$. Let the input data $\mathbf{x} : \Omega \to \mathcal{X}$ be random variables with distributions $P_\mathbf{x}$. We will explore how, from $\mathbf{x}$ and using a neural network, we extract a series of $k$ features. We will assume a classical vision neural network that admits a series of $n$ intermediate spaces, such that:

$$\mathbf{f}_\ell : \mathcal{X} \to \mathcal{A}_\ell \ \text{ with } \ \ell \in \{1, \ldots, n\}.$$

Initially, one might suggest that a feature is a dimension of the model, meaning, for example, that a feature could be a neuron in the last layer of the model $\mathbf{z} = \mathbf{f}_n(\mathbf{x})_i, i \in \{1, \ldots, |\mathcal{A}_n|\}$, thus each of the neurons would be a feature. However, several recent studies [83, 8, 25, 29, 32] have shown that our models actually learn a multitude of features, far more than the number of neurons, which explains, for example, why they are not mono-semantic [77, 19], which could also hinder our study of features. Therefore, we use a recent explainability method, Craft [30], to extract more features

than neurons and avoid this problem of superposition – or feature collapse. With $\boldsymbol{f}_n(\mathbf{x})$ being the penultimate layer, we extract a large number of features, five times more than the number of neurons, using an over-complete dictionary of concepts $\mathbf{D}^\star \in \mathbb{R}^{k \times |\mathcal{A}_n|}$, with $k \gg |\mathcal{A}_n|$. This dictionary is obtained by optimization over the entire training set and contains a total of $k = 10,000$ features. Thus, for a new point $\mathbf{x}$, we obtain the value of the $k$ features – we recall that the number of features is greater than the number of neurons, $k \gg |\mathcal{A}_n|$ – by solving the following optimization problem:

$$\boldsymbol{z} = \arg\min_{\boldsymbol{z} \geq 0} ||\boldsymbol{f}_n(\mathbf{x}) - \boldsymbol{z}\mathbf{D}^\star||_F$$

With $\boldsymbol{z} \in \mathbb{R}^k$ being the value for each feature of the image $\mathbf{x}$, in particular, and from now on for ease of notation, we consider $z_i \in \mathbb{R}, i \in \{1, \dots, k\}$ that we will simply denote $z$ for the rest of the paper, as a *specific feature* for which we want to compute a complexity score. Thus, in our work, a feature refers to a random scalar value extracted by a dictionary learning method on the activations. More details and full derivation regarding the training of $\mathbf{D}^\star$ are available in the Appendix A.

**Complexity through the Lens of Computation.** To formalize this, still on $(\Omega, \mathcal{F}, \mathbb{P})$, we denote the output space $\mathcal{Z}$, and $z : \Omega \to \mathcal{Z}$ are random variables of a feature of interest with distributions $P_z$. The joint random vector $(\mathbf{x}, z)$ representing an image $\mathbf{x}$ and the value of its feature $z$ on $(\Omega, \mathcal{F})$ has a joint distribution $P$ defined over the product space $\mathcal{X} \times \mathcal{Z}$. Furthermore, $\mathcal{P}(\mathcal{Z})$ denotes the set of all probability measures on $\mathcal{Z}$. We can now associate, for an $\mathbf{x}$ which we recall is a real-valued random variable, a corresponding feature $z$, another real-valued random variable, and we seek to correctly evaluate the complexity of the mapping from $\mathbf{x}$ to $z$. For this, we turn to the $\mathcal{V}$-Information [115] that generalizes and extends the classical mutual information $\mathcal{I}(\cdot, \cdot)$ from Shannon's theory by overcoming its inability to take into account the computational capabilities of the decoder. Indeed, for two (not necessarily independent) random variables $\mathbf{x}$ and $z$, and for any bijective mapping $\gamma : \mathcal{X} \to \mathcal{X}$, Shannon's mutual information remains unchanged: $\mathcal{I}(\mathbf{x}, z) = \mathcal{I}(\gamma(\mathbf{x}), z)$.

Consider, for instance, $\gamma$ as a cryptographic function that encrypts an image $\mathbf{x}$ using a bijective key-based algorithm (e.g., the AES encryption algorithm). If $\mathbf{x}$ represents the original image, and $\gamma(\mathbf{x})$ represents the cipherimage, the mutual information between $\mathbf{x}$ and $z$ remains unchanged. This is because the encryption is a bijective process, and the information content is preserved. However, in practice, the encrypted images would be much harder to decode and use for training a model compared to the original one, without access to the decryption key. Another example we may think of is $\gamma$ as a pixel shuffling operation. The information carried by $\mathbf{x}$ does not disappear after processing by $\gamma$. However, it may be harder to extract in practice.

This demonstrates the practical importance of $\mathcal{V}$-Information, as it considers the computational effort required to decode the information, highlighting the difference between *theoretical* and *practical* accessibility of information. Specifically, the $\mathcal{V}$-information proposes taking into account the computational constraint of the decoder by assuming it can only extract information using a *predictive family* $\mathcal{V} \subseteq \mathfrak{F} = \{\eta : \mathcal{X} \cup \{\varnothing\} \to \mathcal{P}(\mathcal{Z})\}$. The authors [115] then define the $\mathcal{V}$-entropy and the $\mathcal{V}$-conditional entropy as follows:

$$H_\mathcal{V}(z) = \inf_{\eta \in \mathcal{V}} \mathbb{E}_{P_z}(-\log \eta(\varnothing; z)), \qquad H_\mathcal{V}(z|\mathbf{x}) = \inf_{\eta \in \mathcal{V}} \mathbb{E}_P(-\log \eta(\mathbf{x}; z)). \qquad (1)$$

Where $\eta(\cdot; \cdot)$ is a function from $\mathcal{X} \cup \{\varnothing\} \to \mathcal{P}(\mathcal{Z})$ that returns a probability density $\eta(\mathbf{x}; \cdot)$ on $\mathcal{Z}$ using side information $\mathbf{x}$, or without side information $\varnothing$. The predictive family $\mathcal{V}$ summarizes the computational capabilities of the decoder. When $\mathcal{V}$ contains all possible functions, $\mathcal{V} = \mathfrak{F}$, it recovers Shannon's entropy as a special case. Intuitively, we seek the best possible prediction for $z$ knowing $\mathbf{x}$ by maximizing the log-likelihood. Continuing, we naturally introduce the $\mathcal{V}$-information:

$$\mathcal{I}_\mathcal{V}(\mathbf{x} \to z) = H_\mathcal{V}(z) - H_\mathcal{V}(z|\mathbf{x}).$$

The complexity of the mapping from $\mathbf{x}$ to $z$ can now be assessed by examining a hierarchy of predictive families $\mathcal{V}_1 \subset \dots \subset \mathcal{V}_n$ of increasing expressiveness, like explored in [61]. Each predictive family $\mathcal{V}_\ell$ corresponds to a partial forward up to depth $\ell$, followed by a decoding step. This involves determining at which point we can decode or make the information from $\mathbf{x}$ to $z$ *available*. Formally, we define the complexity of the feature as dependent of the cumulative $\mathcal{V}$-information across layers:

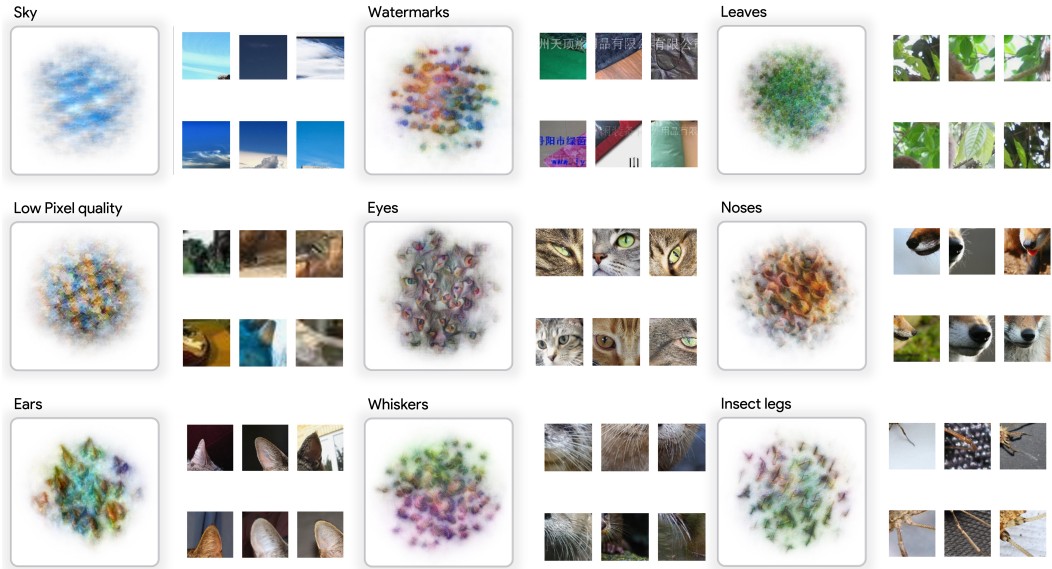

Figure 3: **Visualization of Meta-features, sorted by Complexity.** We use Feature visualization [84, 31] to visualize the Meta-features found after concept extraction. The entire visualization for each Meta-feature can be found in Appendix B.

$$K(\mathbf{z}, \mathbf{x}) = 1 - \frac{1}{n} \sum_{\ell}^{n} \mathcal{I}_{\mathcal{V}}(\boldsymbol{f}_\ell(\mathbf{x}) \to \mathbf{z}). \qquad (2)$$

Here, we define the predictive family $\mathcal{V}$ as a class of linear probes with Gaussian prior. Under this hypothesis, the associated $\mathcal{V}$-information of this class possesses a closed-form solution (see Appendix C), which serves as the basis for our evaluation. A higher score implies that the feature $\mathbf{z}$ is readily accessible and persists throughout the model's layers. Conversely, a lower score suggests that the feature $\mathbf{z}$ is unveiled only at the very end of the model, if at all.

**Assumption.** Crucially, the correctness of the computation of $\mathcal{I}_{\mathcal{V}}(\boldsymbol{f}_\ell(\mathbf{x}) \to \mathbf{z})$ relies on the hypothesis that each layer $f_\ell$ provides the optimal representation for the downstream linear probe $\eta$. In other words, we assume that $\mathcal{I}_{\mathcal{V}}(\boldsymbol{f}_\ell(\mathbf{x}) \to \mathbf{z}) = \mathcal{I}_{\mathcal{V}_\ell}(\mathbf{x} \to \mathbf{z})$, or again that $\eta_\ell^* = \eta^* \circ f_\ell$. This hypothesis is reasonable, since a neural network is essentially "linearizing" the training set—projecting the training set into a space in which it is linearly separable. Thus, it makes sense to assume that each layer attempts to make the feature linearly decodable as efficiently as possible. If this condition is violated, the complexity measure may overestimate the true complexity of a feature (since we can only underestimate the $\mathcal{V}$-information). For example, this may happen if the optimal path to calculate a feature requires deviating from the linear decoding to make it easier to decode later. While some recent works have motivated a slightly different complexity metric based on redundancy [24], we show in Appendix E that our complexity measure is inherently linked to redundancy.

## 3  *What* Do Complex Features Look Like? A Qualitative Analysis

This section presents a qualitative investigation of relatively simple versus more complex features. Drawing from critical insights of recent studies, which indicate a tendency of neural networks to prefer input features that are both predictive and not overly complex [48], this analysis aims to better understand the nature of features that are easily processed by models versus those that pose more significant challenges. Indeed, understanding the types of features that are too complex for our model can help us anticipate the types of shortcuts the model might rely on and, on the other hand, design methods to simplify the learning of complex features. This section of the manuscript is intentionally qualitative and aims to be exploratory. We applied our complexity metric to 10,000 features extracted from a fully trained ResNet50. For each feature, we computed the complexity score $K(\mathbf{z}, \mathbf{x})$ using a subset of 20,000 images from the validation set. Recognizing the impracticality of manually examining each of the 10,000 features, we employed a strategy to aggregate these features into a more manageable number of groups that we called Meta-features.

**Method for Aggregating Features into Meta-features.** To condense the vast array of features into a reduced number of similar features, we applied K-means clustering to the feature dictionary $\mathbf{D}^\star$, resulting in 150 distinct clusters. These clusters represent collections of features, referred to as Meta-features $\mathcal{C} = \{\boldsymbol{v}_1, \ldots, \boldsymbol{v}_{|\mathcal{C}|}\}$; we then computed an average complexity score for each group. By selecting a diverse range of 30 clusters, chosen to cover a spectrum of complexity levels from the simplest to the most complex features, we aimed to provide a comprehensive overview of the diversity of feature complexity within the model. We propose to visualize the distance matrix in $\mathbf{D}^\star$, showing feature complexity in Figure 2. This approach offers preliminary insights into features seen as simple or complex by the model.

**Simple Features.** Among the simpler features, we find elements primarily based on color, such as *sky* and *sea*, as well as simple pattern detectors like *line* detectors and low-frequency detectors exemplified by *bokeh*. Interestingly, features geared towards text detection, such as *watermark*, are also included in this group. These findings align with previous studies [117, 96, 12, 82], which have shown that neural networks tend to identify color and simple geometric patterns in the early layers as well as low-frequency detectors. This suggests that these features are relatively easy for neural networks to process and recognize. Furthermore, our findings detailed in Appendix 11 corroborate the theoretical work posited in [11, 72]: robust learning possibly induces the learning of shortcuts or reliance on "easy" features within the model.

**Medium Complexity Features.** Features with medium complexity reveal more nuanced and sometimes unexpected characteristics. We find, for example, *low-quality* detectors sensitive to low-resolution images. Additionally, a significant number of concepts related to *human elements* were observed despite the absence of a dedicated *human* class in ImageNet. *Trademark-related* features, distinct from simpler *watermark* detectors, also reside within this intermediate complexity bracket.

**Complex Features.** Among the most complex features, we find several Meta-features that exhibit a notable degree of structural coherence, including categories such as *insect legs*, *curves*, and *ears*. These patterns represent structured configurations that are ostensibly more challenging for models to process than more localized features, echoing the ongoing discussion about texture bias in current models [10, 37, 46]. Intriguingly, the most complex Meta-features identified, namely *whiskers* and *insect legs*, embody types of filament-like structures. Interestingly, we note that those types of features are known to be challenging for current models to identify accurately [60], aligning with documented difficulties in path-tracking tasks [66]. Such tasks have revealed current models' limitations in tracing paths, which parallels challenges in connectomics [89], particularly in filament segmentation—a domain recognized for its complexity within deep learning research.

Now that we've browsed simple and complex features, another question arises: how does the model build these features during the forward pass? For instance, **where** within the model does the formation of a watermark detector feature occur? And for more complex features that require greater structure, in which block of computation are these features formed within the model?

## 4 *Where* do Complex Features Emerge

As suggested by previous work, simple features, like color detectors and low-frequency detectors, may already exist within the early layers of the model. An intriguing question arises: how does the model ensure the propagation of these features to the final latent space $\boldsymbol{f}_n$, where features are extracted? A key component to consider in addressing this question is the role of residual connections within the ResNet [45] architecture. The formulation of a residual connection in ResNet blocks is mathematically represented as:

$$\boldsymbol{f}_{\ell+1}(\mathbf{x}) = \underbrace{\boldsymbol{f}_\ell(\mathbf{x})}_{\text{"Residual" branch}} + \underbrace{(\boldsymbol{g}_\ell \circ \boldsymbol{f}_\ell)(\mathbf{x})}_{\text{"Main" branch}}$$

This equation highlights two distinct paths: the "Residual" branch, which facilitates the direct

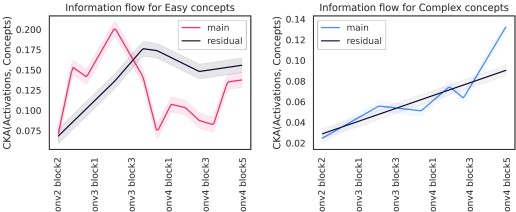

Figure 4: **Simple Features Teleported by Residuals. (Left)** CKA between residual branch activations $\boldsymbol{f}_\ell$ and final concept value $z$. For simple concepts, beyond a certain layer (block 3), the residual already carries nearly all the information, effectively teleporting it to the last layer. **(Right)** Conversely, for complex features, both the main and residual branches gradually construct the features during the forward pass.

transfer of features from $\boldsymbol{f}_\ell$ to the subsequent layer $\ell + 1$, and the "Main" branch, which introduces additional transformations to $\boldsymbol{f}_\ell$ through additional computation $\boldsymbol{g}_\ell$ to enhance its representational capacity. We aim to investigate the *flow* of simple and complex features through these branches. In our analysis, we examine two subsets of features: 100 features of the highest complexity (top-1 percentile) and 100 features of the lowest complexity (bottom-1 percentile). We measure the Centered Kernel Alignment (CKA) [57] between the final concept values $z$ and the activations from (A) the "Residual" branch $\boldsymbol{f}_\ell$, and (B) the "Main" branch ($\boldsymbol{g}_\ell \circ \boldsymbol{f}_\ell$), at each residual block, as a proxy for concept information contained in each branch. The findings, illustrated in Figure 4, reveal that simple features are efficiently "teleported" to later layers through the residual branches – in other words, once computed, they are passed forward with little subsequent modification. In contrast, complex concepts are incrementally built up through an interactive process involving the "main" and "residual" branches. This understanding of feature evolution within network architectures emphasizes the importance of residual connections. This insight, though expected, clarifies a common conception by showing that simple features utilize the residual branch. The next step is to examine the temporal dynamics of feature development, specifically investigating when complex and simple concepts emerge during model training.

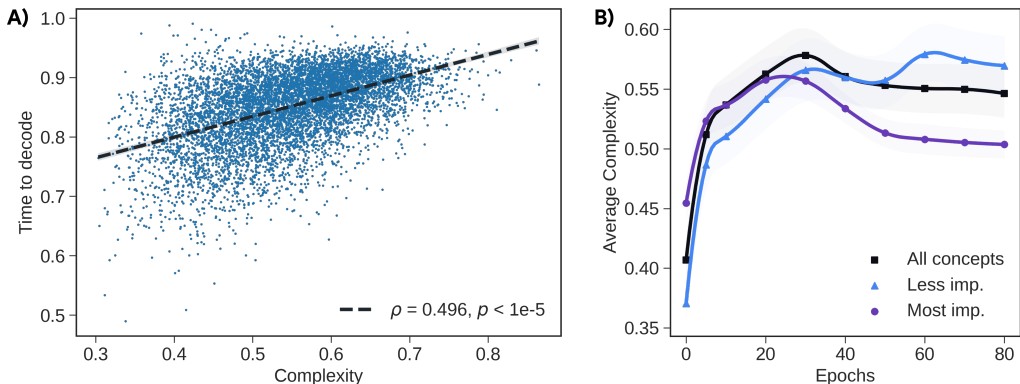

Figure 5: **A) Complex features emerge later in training.** There is a strong correlation between the complexity of a feature and the requisite temporal span for its decoding. The temporal decoding score, $\Lambda$, is derived as the mean $\mathcal{V}$-information across epochs, with $\mathcal{V}$ representing the class encompassing linear models. A low score indicates a feature is accessible earlier during the training continuum, whereas a high score implies its tardy availability. The correlation between these scores suggests that complex features tend to emerge later in training. **B) Important features are being compressed by the neural network: *Levin Machine* hypothesis.** The average complexity of 10,000 features extracted independently at each epoch increases rapidly before stabilizing (the black curve shows the average). However, among the top-1% of features in terms of importance, complexity decreases over time, as if the model is self-compressing or simplifying, akin to a sedimentation process.

## 5   *When* do Complex Features Arise

Figure 1 raises an important question: Does the complexity of a feature influence the time it takes to develop during training? To explore this, we refer to the 10,000 features extracted at the final epoch of our model as $\boldsymbol{f}_n^{(e)}$, and we use $\boldsymbol{f}_n^{(i)}$ to represent the penultimate layer of the model at any given epoch $i$, where $i \in \{1, \ldots, e\}$ and $e$ represents the total number of epochs. We aim to determine how early each feature can be detected in previous epochs $\boldsymbol{f}_n^{(i)}$ for $i < e$. This involves calculating a specific decoding score; in our scenario, we define this score as $\mathcal{I}_\mathcal{V}$—the measure of $\mathcal{V}$-information between the model's penultimate activations across epochs and an ultimate feature values, where $\mathcal{V}$ is the set of linear models. This metric helps us assess whether a feature was "readily available" at a certain epoch $i$. The cumulative score $\Lambda$ is calculated by averaging this measure across all epochs, leading to our score:

$$\Lambda(\mathbf{x}, z) = 1 - \frac{1}{e} \sum_i^e \mathcal{I}_\mathcal{V}(\boldsymbol{f}_n^{(i)}(\mathbf{x}) \to z).$$

The results, as illustrated in Figure 5A, showcase the complexity of a feature ($K$) with it's Time to Decode ($\Lambda$) score. An observed correlation coefficient nearing $0.5$ intimates that features of heightened complexity are generally decoded later during the training epoch. This finding suggests a nuanced interrelation between the layer for which a feature is available and the epoch of discovery: a feature decoded later in the forward pass trajectory also came online later in training. This naturally leads us to the question of the dynamics of model training. Can we get a deeper understanding of how precisely complex concepts are formed within the model? Does the model develop complex features solely upon necessity, thereby suggesting a correlation between the complexity of a feature and its importance?

## 6   Complexity and Importance: A Subtle Tango

Numerous studies have proposed hypotheses regarding the relationship between the importance and complexity of features within neural networks. A particularly notable hypothesis is the simplicity bias [3, 51, 110], which suggests that models leverage simpler features more frequently. This section aims to quantitatively validate these claims using our complexity metric paired with the importance of each feature. Because features are extracted from the penultimate layer, a closed-form relationship between features and logits can be derived due to the linear nature of this relationship. By analyzing this relationship over training for features of different complexity, we identify a surprising novel perspective: models appear to *reduce* the complexity of their important features. This process is analogous to sedimentation and mirrors the operation of a *Levin* Universal Search [63]. The model incrementally shifts significant features to earlier layers, taking time to identify simpler algorithms in the process.

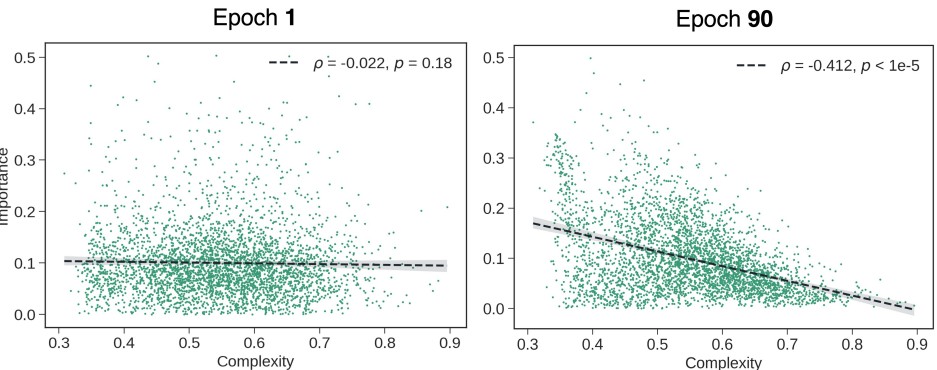

Figure 6: **Simplicity bias appears during training.** Complexity vs. Importance of 10,000 features extracted from a ResNet50 at Epochs 1 and 90 of training. In Epoch 1, important features are not necessarily simple and seem uniformly distributed. In contrast, by the end of training, there is a clear simplicity bias, consistent with numerous studies: the model prefers to rely on simpler features.

**Importance Measure.** The feature extraction framework outlined in Section 2 offers a structured approach to estimating the importance of a feature within the network. Specifically, the feature vector $z \in \mathbb{R}^k$ is linearly related to the model's decision-making process, exemplified by a logit calculation $y = z\mathbf{D}^\star W \in \mathbb{R}$, where $W \in \mathbb{R}^{|\mathcal{A}_n|}$ represents the weights of the penultimate layer for the class-specific logit. The contribution of the $i$-th feature, $z_i$, to the logit $y$ can be precisely measured by leveraging the gradient-input formulation, which is optimal for fidelity metrics within a linear context [5, 32]. This optimality and the closed-form expression are feasible primarily because the analysis is confined to the penultimate layer of the network. Formally, the importance of a feature $z_i$ is defined as: $\mathbf{\Gamma}(z_i) = \mathbb{E}_{\mathbb{P}_z}\left(||\frac{\partial y}{\partial z_i} \cdot z_i||\right)$. In essence, the importance measure $\mathbf{\Gamma}(z_i)$ quantifies the average contribution of the $i$-th feature to the class-specific logit – essentially, the average score that each feature brings to the decision logit. More details on importance measures and the effect of inhibition features are available in Appendix G.

**Models Prefer Simple Features.** The analysis, supported by Figure 6 (right), demonstrates a clear trend indicating the model's simplicity bias. Among the 10,000 features extracted in the final epoch,

more complex features—characterized by higher $K(\mathbf{x}, z)$ values—are generally assigned lower importance ($\mathbf{\Gamma}(z)$). In contrast, simpler features predominantly influence the model's decisions. The plot on the left showcases the complexity and importance of 10,000 concepts extracted at the end of the first epoch; we observe that the model does not exhibit this simplicity bias at the end of the first epoch. More detail and study on the role of complex concept is proposed in Appendix D. This observation raises the question of the dynamic interplay between feature complexity and importance. To further investigate, we did a detailed analysis of the evolution of feature complexity and importance throughout the training process.

**Model as *Levin's Machine*: Simplifying the Complexity of Important Features.** A closer examination of the evolution of feature importance over time reveals an interesting phenomenon in Figure 5B: the emergence of two distinct phases during training. Initially, there is a global increase in feature complexity, with the model beginning its training with relatively simple features. Surprisingly, this is followed by a phase where the model actively reduces its overall complexity, specifically targeting and simplifying its most important features. The model appears to be "shortening" the computational "programs" responsible for generating these significant features. This observation suggests that the ResNet50 under study, like a Levin Machine, develops simpler computational paths for crucial features. Put simply, our complexity metric shows that important features are extracted at earlier layers, resembling sedimentation with foundational elements near the network's input.

This behavior presents a novel perspective on how neural networks might be intrinsically driven to generalize by simplifying the computation graph of their important features. However, at least at the early stages of learning, it also challenges our assumption that each layer is optimized to provide a linearly-separable representation for the downstream linear probe – early in learning, this assumption is clearly violated since some complex features could be represented more simply than they are initially. Thus, future work will be needed to fully disentangle the interaction of complexity and importance over training.

## 7   Conclusion

We introduced a complexity metric for neural network features, identifying both simple and complex types. We have shown where simple features flow – through residual connections – as opposed to complex ones that develop via collaboration with main branches. Our study further revealed that complex features are learned later in training than simple ones. We have concluded by exploring the relationship between feature complexity and importance, and discovered that the simplicity bias found in neural networks becomes more pronounced as training progresses. Surprisingly, we found that important features simplify over time, suggesting a *sedimentation process* within neural networks that compresses important features to be accessible earlier in the network.

## Acknowledgments

We thank Amal Rannen-Triki and Noah Fiedel for feedback on the manuscript, and Amal, Robert Geirhos, Olivia Wiles, and Mike Mozer for interesting discussions.

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

# A  Feature Extraction

**Dictionary Learning.**   To comprehensively analyze the complexity of features extracted from a deep learning model, we employed a detailed feature extraction process using dictionary learning, specifically utilizing an over-complete dictionary. This approach allows each activation $\boldsymbol{f}_n(\boldsymbol{x}) \in \mathcal{A}_\ell$ to be expressed as a linear combination of multiple basis elements (direction, also called atoms) $\boldsymbol{d} \in \mathcal{A}_\ell$ from the dictionary $\mathbf{D}^\star = \{\boldsymbol{d}_1, \ldots, \boldsymbol{d}_k\}$ coupled with some sparse coefficient $\boldsymbol{z} \in \mathbb{R}^k$ associated to each atoms.

The over-completness of $\mathbf{D}^\star$ means that the dimension of the dictionnary ($k$) is larger than the dimension of the activations space $k >> |\mathcal{A}_\ell|$. This property allow us to overcome the superposition problem [29] essentially stating that there be more feature than neurons.

Mathematically, given an activation function $\boldsymbol{f}_n(x)$, it can be represented as a linear combination of atoms from the dictionary $\mathbf{D}$, expressed as:

$$\boldsymbol{f}_n(\boldsymbol{x}) \approx \boldsymbol{z}\mathbf{D}^\star = \sum_i^k z_i \boldsymbol{d}_i$$

where $z_i$ are the coefficients indicating the contribution of each atom $\boldsymbol{d}_i$ from the dictionary.

**Implementation.**   Our implementation was inspired by Craft [30], leveraging the properties of ReLU activations in ResNet50. Given that ReLUs induce non-negativity of the activation, we employed Non-Negative Matrix Factorization (NMF) [59, 113] for the reconstruction, as it naturally aligns with the sparsity and non-negativity constraints of ReLU activations. Unlike PCA, which cannot produce overcomplete dictionaries and may result in non-positive activations, NMF can create overcomplete dictionaries in this context.

The dictionary $\mathbf{D}^\star$ was trained to reconstruct the activations $\boldsymbol{f}_n(\boldsymbol{x})$ using the entire ImageNet training dataset, comprising 1.2 million images. Formally, for the set of images $\boldsymbol{X}$ and their corresponding activations $\boldsymbol{f}_n(\boldsymbol{X})$, the objective was to minimize the reconstruction error:

$$||\boldsymbol{f}_n(\boldsymbol{X}) - \boldsymbol{Z}\mathbf{D}^\star||_F,$$

ensuring that $\boldsymbol{f}_n(\boldsymbol{X})$ can be closely approximated by $\boldsymbol{Z}\mathbf{D}^\star$. Additionally, the NMF framework enforces non-negativity constraints on the dictionary matrix $\mathbf{D}^\star \geq 0$ and the coefficients $\boldsymbol{Z} \geq 0$:

$$(\boldsymbol{Z}, \mathbf{D}^\star) = \underset{\boldsymbol{Z} \geq 0, \mathbf{D}^\star \geq 0}{\arg\min} ||\boldsymbol{f}_n(\boldsymbol{X}) - \boldsymbol{Z}\mathbf{D}^\star||_F.$$

The dictionary $\mathbf{D}^\star$ was designed to encapsulate 10 concepts per class, resulting in a total of 10,000 concepts. To augment the training samples for NMF, we exploited the spatial dimensions of the last layer of ResNet50, which has 2048 channels with a spatial resolution of 7x7. By training the NMF independently on each of the 49 spatial dimensions, we effectively increased the number of training samples to approximately 58 million artificial samples (channel activations).

We utilized the block coordinate descent solver from Scikit-learn [86] to solve the NMF problem. This algorithm decomposes the problem into smaller subproblems, making it more tractable. The optimization process continued until convergence was achieved with a tolerance of $\varepsilon = 10^{-4}$, ensuring the dictionary was sufficiently optimized for accurate feature extraction. Post-training, the reconstructed activations $\boldsymbol{Z}\mathbf{D}^\star$ retained over 99% accuracy in common predictions compared to the original activations $\boldsymbol{f}_n(\boldsymbol{X})$.

**Extracting Features for New Data Points.**   Once the dictionary $\mathbf{D}^\star$ was trained, it was fixed. For any new input $\boldsymbol{x}$, the corresponding feature $\boldsymbol{z}$ was extracted by solving a Non-Negative Least Squares (NNLS) problem. This mapping of new input activations $\boldsymbol{f}_n(\boldsymbol{x})$ to the learned feature space was performed by minimizing the following objective:

$$\boldsymbol{z} = \underset{\boldsymbol{z} \geq 0}{\arg\min} ||\boldsymbol{f}_n(\boldsymbol{x}) - \boldsymbol{z}\mathbf{D}^\star||_F.$$

This optimization problem is convex, ensuring computational feasibility and robust feature extraction for new data points.

# B  Visualization of Meta-features

To visualize each feature, we used feature visualization methods [84]. Specifically, a recent improvement [31] re-parameterizes the original feature visualization optimization problem—i.e., finding an image $x$ that maximizes a direction, channel, or neuron—by optimizing only the phase of a Fourier buffer while fixing the image magnitude. This approach produces more realistic images and prevents high-frequency leakage that results in adversarial positives.

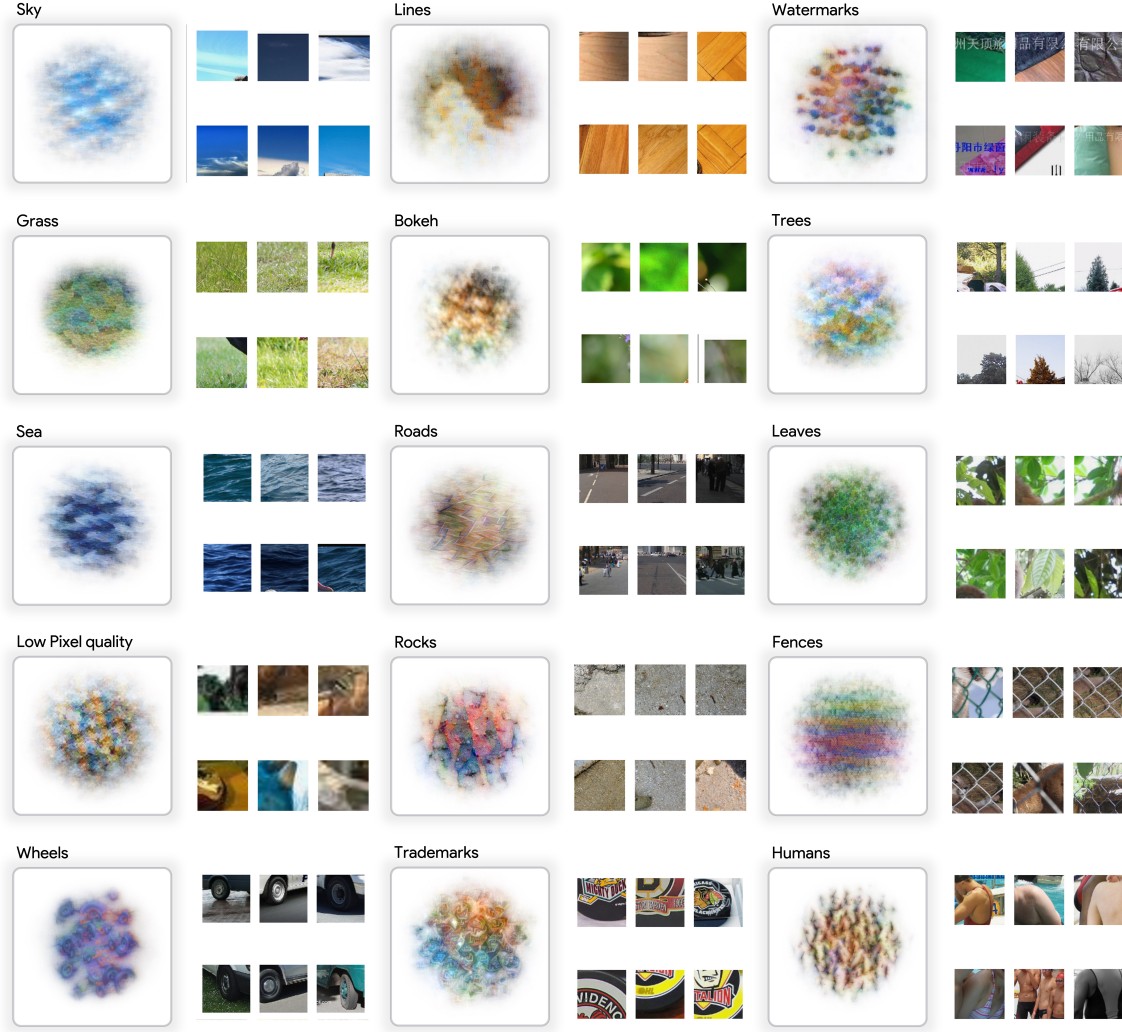

Figure 7: **Visualization of Meta-Features, sorted by Complexity.** Feature visualization using MACO [31] for the most simple (1-15) of the 30  Meta-features found on the 10,000 features extracted.

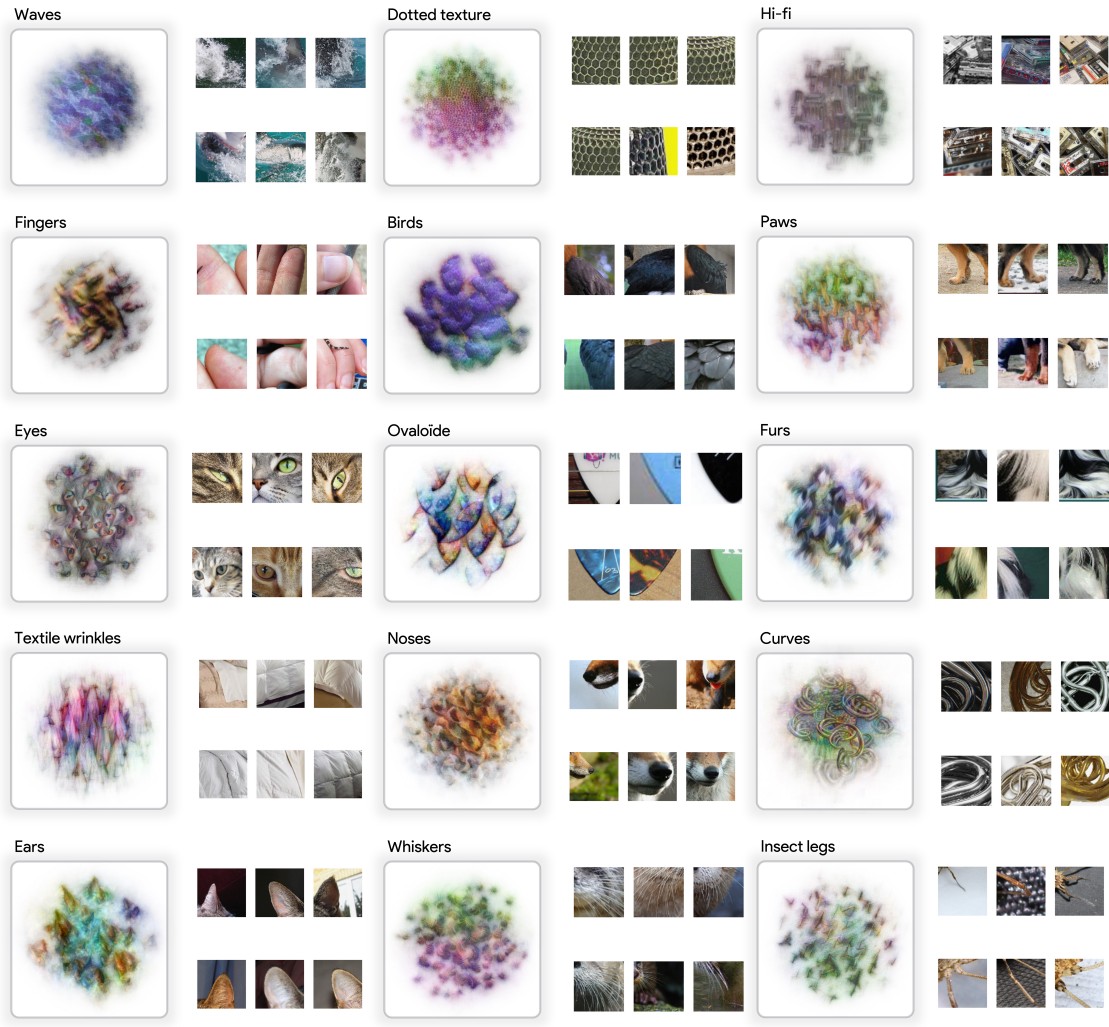

Figure 8: **Visualization of Meta-Features, sorted by Complexity.** Feature visualization using MACO [31] for for the most complex (15-30) of the 30  Meta-features found on the 10,000 features extracted.

# C   Complexity measure

In this section, we detail the closed-form expression of the $\mathcal{V}$-information when the predictive family $\mathcal{V}$ consists of linear classifiers with Gaussian posteriors. Specifically, $\mathcal{V}$ is defined as follows:

$$\mathcal{V} = \begin{cases} \eta : \mathbf{x} \to \mathcal{N}(\psi(\mathbf{x}), \sigma^2), \text{ with } \mathbf{x} \in \mathcal{X} \text{ and } \psi \in \Psi; \\ \varnothing \to \mathcal{N}(\mu, \sigma^2), \text{ with } \mu \in \mathbb{R}, \sigma^2 = \frac{1}{2}; \end{cases}$$

where $\Psi = \{\mathbf{x} \mapsto M\mathbf{x} \mid M \in \mathbb{R}^d\}$ is a set of linear predictors. This setting corresponds to the linear decoding we apply during the computation of $\mathcal{V}$-information. In this context, a closed-form solution is available (see [115]):

$$\begin{aligned}
\mathcal{I}_\mathcal{V}(\mathbf{x} \to z) &= H_\mathcal{V}(z) - H_\mathcal{V}(z \mid \mathbf{x}) \\
&= \inf_{\mu \in \mathbb{R}} \mathbb{E}_{z \sim P_z} \left[ -\log \frac{1}{\sqrt{2\pi\sigma^2}} e^{-\frac{(z-\mu)^2}{2\sigma^2}} \right] - \inf_{\psi \in \Psi} \mathbb{E}_{\mathbf{x}, z \sim P} \left[ -\log \frac{1}{\sqrt{2\pi\sigma^2}} e^{-\frac{(z-\psi(\mathbf{x}))^2}{2\sigma^2}} \right] \\
&= \inf_{\mu \in \mathbb{R}} \mathbb{E}_{z \sim P_z} \left[ \frac{(z-\mu)^2}{2\sigma^2} \right] - \inf_{\psi \in \Psi} \mathbb{E}_{\mathbf{x}, z \sim P} \left[ \frac{(z-\psi(\mathbf{x}))^2}{2\sigma^2} \right] \\
&= \frac{1}{2\sigma^2} \left( \inf_{\mu \in \mathbb{R}} \mathbb{E}_{z \sim P_z} \left[ (z-\mu)^2 \right] - \inf_{\psi \in \Psi} \mathbb{E}_{\mathbf{x}, z \sim P} \left[ (z-\psi(\mathbf{x}))^2 \right] \right) \\
&= \frac{\text{Var}(z)}{2\sigma^2} \left( 1 - \frac{\inf_{\psi \in \Psi} \mathbb{E}_{\mathbf{x}, z \sim P} \left[ (z-\psi(\mathbf{x}))^2 \right]}{\text{Var}(z)} \right) \\
&= \frac{\text{Var}(z)}{2\sigma^2} R^2 \\
&= \text{Var}(z) R^2.
\end{aligned}$$

Here, $R^2$ is the coefficient of determination. Therefore, the following inequalities hold:

$$0 \leq \mathcal{I}_\mathcal{V}(\mathbf{x} \to z) \leq \text{Var}(z).$$

Given that the input data are centered and scaled, we typically have $\text{Var}(z)$ around 1 (or less in case of Layer normalization). Furthermore, residual connections and batch normalization tend to preserve this scaling in deeper layers. We note, however, that our score $K(z, \mathbf{x})$ is not strictly bounded. Indeed, we define complexity as the opposite of the average $\mathcal{V}$-information across layers: complex features are those that are harder to decode. We add a shift of 1 for the ease of plotting. Empirically, we observed that this adjustment yields $K(z, \mathbf{x})$ in the range $[0, 1]$, with 1 indicating a complex feature that is not available and 0 indicating a simple feature that is fully available.

# D  Feature Support Theory

While simplifying important features underscores a trend toward computational efficiency, the role of complex features within the model deserves a closer examination. Despite these features often being deemed less important directly, they contribute significantly to the model's overall performance, a paradox that can lead us to introduce the concept of "support features." These are a set of features that may not carry substantial importance individually, but that collectively play a crucial role in the model's decision-making process.

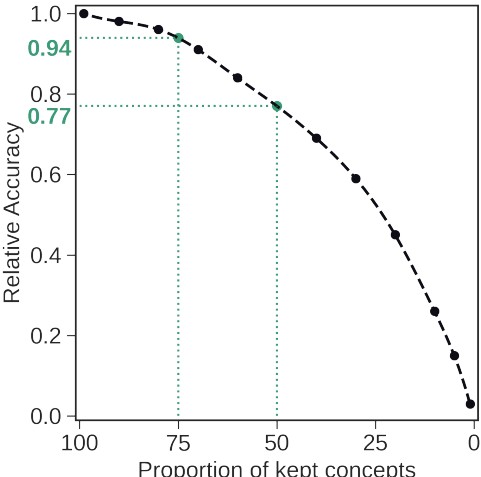

Figure 9: **"Support Features" hypothesis.** The majority of complex features are not very important, but play a non-negligible role and contribute to significant performance gains. This paradox is referred to as the "support features," a large ensemble of features individually of little to very little importance to the model but collectively holding a significant role.

The presence of numerous complex features, whose importance on average is less pronounced, poses a conundrum. However, these features are far from redundant. Experiments conducted by progressively removing the most complex concepts from the model demonstrate a noticeable impact on performance, as illustrated in Figure 9. This empirical evidence supports the theory that, while individually, these complex features may not be pivotal, their collective presence contributes indispensably to the robustness and adaptability of the model. These results are reminiscent of prior findings that low-importance model components that are removed in pruning may nevertheless contribute to model accuracy on rare items [50].

This observation aligns with the broader understanding of neural network functionality, where diversity in feature representation—spanning from simple to complex—enhances the model's ability to generalize and perform across varied datasets and tasks. Therefore, the "Feature Support Theory" underscores an essential aspect of neural network design and training: integrating and preserving a wide spectrum of features, regardless of their individual perceived importance, are vital for achieving high levels of performance and robustness.

# E   Complexity and Redundancy

To further understand the link between feature complexity and redundancy, we utilized the redundancy measure from [76]. Our findings indicate that complex features tend to be less redundant, as depicted in Figure 10. This observation aligns with the strong correlation between our complexity measure and the redundancy-based complexity measure proposed by [24].

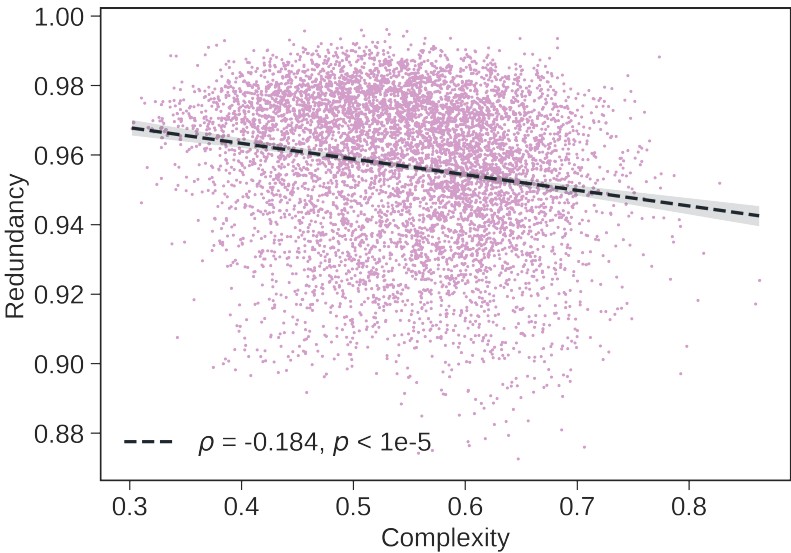

Figure 10: **Complex features are less redundant.** Using the redundancy measure from [76], we show that our complex features tend to be less redundant. This result also confirms a link between our complexity measure and the one recently proposed by [24], which is also based on redundancy.

To quantify redundancy, [76] employed a modified version of Centered Kernel Alignment (CKA) [57], a measure of similarity between two sets of activation features. We briefly recall that CKA between two set of activations $\boldsymbol{A}, \boldsymbol{B}$ in $\mathbb{R}^d$ is defined as follows:

$$\mathrm{CKA}(\boldsymbol{A}, \boldsymbol{B}) = \frac{\|\boldsymbol{K_A K_B}\|_F^2}{\|\boldsymbol{K_A K_A}\|_F \|\boldsymbol{K_B K_B}\|_F}$$

where $\boldsymbol{K_A}$ and $\boldsymbol{K_B}$ are the Gram matrices of the feature activations $\boldsymbol{A}$ and $\boldsymbol{B}$, respectively, and $\|\cdot\|_F$ denotes the Frobenius norm.

In our analysis, we calculated the CKA measure between a feature $z$ and the activations for a set of 2,000 images from the validation set $\boldsymbol{f}_n(\boldsymbol{X})$. Subsequently, we compared this with the CKA measure when a portion of the activation is masked using a binary mask $\boldsymbol{m} \in \{0, 1\}^{|\mathcal{A}_\ell|}$, denoted as $\mathrm{CKA}(z, \boldsymbol{f}_n(\boldsymbol{X}) \odot \boldsymbol{m})$, where $\odot$ represents element-wise multiplication (Hadamard product). This comparison enabled us to assess whether masking a subset of neurons impacts the decoding of the features. Specifically, to evaluate redundancy, we employed a progressive masking strategy, successively masking 10%, 50%, and 90% of the activation. If the masked activations retain a high CKA with $z$, it indicates that the information remains largely intact, suggesting that the feature is redundantly stored across multiple neurons, sign of a redundant encoding mechanism within the network. Conversely, if masking results in a substantial decrease in CKA, it implies that the information was predominantly localized on a specific neuron. In this scenario, the feature is not redundantly encoded but rather concentrated in specific neurons. This concentration indicates a lower degree of redundancy, as the loss of these specific neurons (throught the masking) leads to a significant reduction in the CKA score.

The final score of redundancy is then the average CKA difference between the original activation and the masked activations:

$$\text{Redundancy} = \frac{\mathbb{E}_{\boldsymbol{m}}\big(\text{CKA}(\boldsymbol{f}_n(\boldsymbol{X}) \odot \boldsymbol{m}, \boldsymbol{z})\big)}{\text{CKA}(\boldsymbol{f}_n(\boldsymbol{X}), \boldsymbol{z})}$$

And averaged across the different level of masking. A high score (1) indicating a high redundancy – i.e. the CKA between the masked activation and with the original activation is similar – while a low score indicate a more localized and thus a lower degree of redundancy.

In summary, our results, as depicted in Figure 10 support the idea that complex features exhibit lower redundancy.

# F Complexity and Robustness

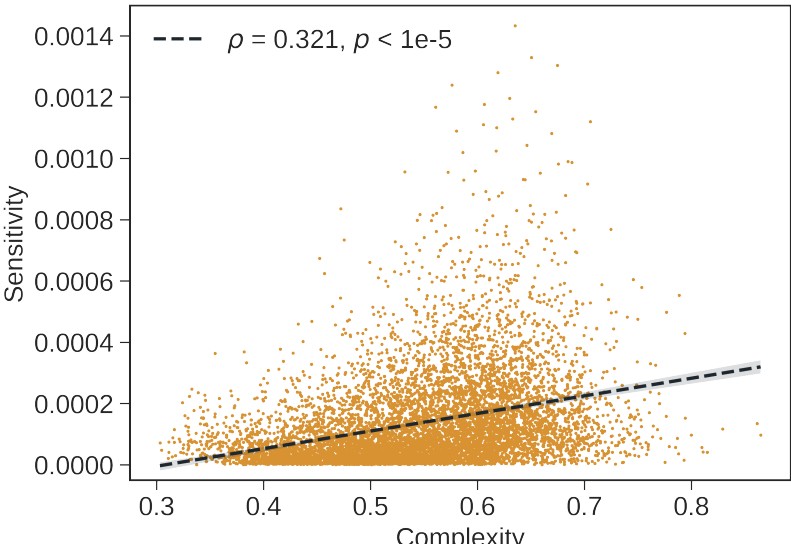

Figure 11: **Complex features are less robust.** This figure illustrates the relationship between feature complexity and robustness, quantified as the variance of the feature value when the image is perturbed with Gaussian noise. The results indicate that more complex features tend to exhibit lower robustness.

To measure robustness, we evaluate the stability of feature responses under perturbations. For each input point $\mathbf{x}$, we add isotropic Gaussian noise with varying levels of standard deviation $\sigma$. The robustness score is determined by measuring the variance in the feature response due to the noise. Formally, let $z(\mathbf{x})$ represent the feature response for input $\mathbf{x}$. We define the perturbed input as: $\tilde{\mathbf{x}} = \mathbf{x} + \mathcal{N}(0, \sigma^2 \mathbf{I})$ where $\mathcal{N}(0, \sigma^2 \mathbf{I})$ represents Gaussian noise with mean 0 and variance $\sigma^2$. The sensitivity score Sensitivity$(z)$ for a feature $z$ is given by:

$$\text{Sensitivity}(z) = \text{Var}(z(\tilde{\mathbf{x}}))$$

Specifically, we sample 100 random noise and repeat this for 3 levels of noise $\sigma \in \{0.01, 0.1, 0.5\}$ to compute the variance in feature response for each input from 2,000 samples from the Validation set of ImageNet to get a distribution of feature value. We also consider other metrics such as the range (min-max) of the feature response, but all methods consistently indicate that more complex features are less robust.

In summary, our results, as shown in Figure 11, demonstrate that complex features exhibit lower robustness. This indicates that features with higher complexity are more sensitive to perturbations and noise, resulting in greater variability in their responses.

# G   Importance Measure

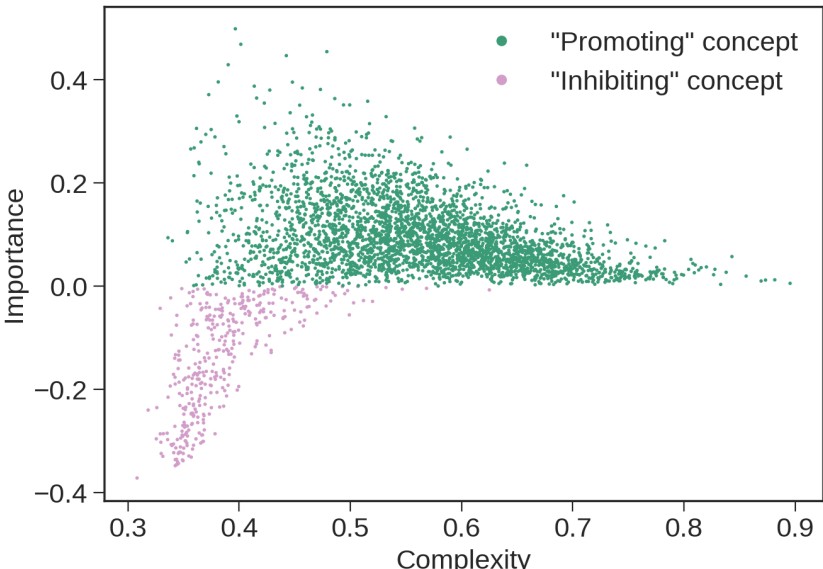

Figure 12: **Inhibiting and non-inhibiting features vs complexity.** Important features can be significant either by inhibition, i.e., removing information from a class, or by adding information for a given class. Each point represents a feature, and violet-colored features generally act as inhibitors ($\mathbf{\Gamma}(z_i) < 0$).

The problem of estimating feature importance is closely related to attribution methods [117, 33, 88, 80, 15, 102, 106, 23], which aim to identify the important pixels for a decision. A recent study has shown that all attribution methods can be extended in the space of concepts [32]. In our case, the features are extracted from the penultimate layer, where the relationship between feature values and logits is linear. We will elaborate on this and demonstrate that the notion of importance in the linear case is easier and optimal methods to estimate importance exist.

**Setup.**   Recall that for a point $\mathbf{x}$, we can obtain its $k$ feature values by solving an NNLS problem $\mathbf{z} = \arg\min ||\mathbf{f}_n(\mathbf{x}) - \mathbf{z}\mathbf{D}^\star||_F$. The vector $\mathbf{z}$ contains the $k$ features in $\mathbb{R}^k$. We can replace the activation of the penultimate layer $\mathbf{f}_n(\mathbf{x})$ with its feature representation in the over-complete basis $\mathbf{z}\mathbf{D}^\star \approx \mathbf{f}_n(\mathbf{x})$. Since we are in the penultimate layer, the model's decision, i.e., the logit $\mathsf{y} \in \mathbb{R}$ for the predicted class, is linearly related to each feature $\mathsf{z}$ by the last weight matrix, denoted as $\mathbf{W}$, as follows:

$$\mathsf{y} = \mathbf{f}_n(\mathbf{x})\mathbf{W} \tag{3}$$
$$\approx \mathbf{z}\mathbf{D}^\star\mathbf{W} \quad \text{with} \quad \mathbf{z} = \arg\min ||\mathbf{f}_n(\mathbf{x}) - \mathbf{z}\mathbf{D}^\star||_F \tag{4}$$
$$= \mathbf{z}\mathbf{W}' \quad \text{with} \quad \mathbf{W}' = \mathbf{D}^\star\mathbf{W} \in \mathbb{R}^k \tag{5}$$

Thus, the energy contributed to the logit by feature $i$ can be directly measured by $\mathbf{W}'_i z_i$ and $\mathsf{y} = \sum_i^k \mathbf{W}'_i z_i$. Consequently, the contribution of a feature $z_i$ can be measured using gradient-input, $(\nabla_{\mathbf{z}}\mathsf{y}) \odot \mathbf{z}$. Several studies [5, 32] have detailed the linear case and shown the optimality of gradient-input with respect to fidelity metrics. They also demonstrated that many methods in the linear case boil down to Gradient-Input, including Occlusion [117], Rise[88], and Integrated Gradient[106].

In our case, we measured the importance by taking the absolute value of the importance vector, i.e., $\mathbf{\Gamma}(z_i) = \mathbb{E}_{\mathbb{P}_z}\left(||\frac{\partial \mathsf{y}}{\partial z_i} \cdot z_i||\right)$. It is natural to question whether this approach might overlook important features due to their inhibitory effects. Indeed, as depicted in Figure 12, numerous features may be important not because they add positive energy to the logits, but by inhibition, i.e., by suppressing

class information. Although this does not alter the implications of our previous observations, it is noteworthy that the majority of inhibitory features are also simple features.

**Prevalence and Importance.** Another property of importance is its close relationship with prevalence [32], which indicates that a frequently occurring feature will, on average, be more important given the same importance coefficient ($\nabla_z y$). In our study, this implies that if the most important features are reduced, these important features are also potentially more frequently present. Consequently, the prevalence of a feature can be a factor explaining this sedimentation process. We refer the reader to a concurrent study that proposed to investigate more deeply this phenomena using a controlled dataset [58].

# H  Features Clustering

The visualization in Figure 2 prompts an important question: are features in our model clustered based on their complexity? Specifically, are there regions in the feature space that are generally more "complex" than others, or that respond primarily to more complex stimuli? Our access to 10,000 features via overcomplete decomposition enables a more detailed analysis compared to traditional neuron-wise studies. We aim to explore three main hypotheses:

- **Hypothesis 1: Features cluster by super-class.** This hypothesis posits that features corresponding to semantically related categories are spatially grouped within the feature space—e.g., concepts related to the dog class are closer to those of the cat class than to unrelated classes like furniture.

- **Hypothesis 2: Features cluster by complexity.** We suggest that features may organize themselves based on their complexity, with simpler features forming distinct clusters separate from more complex ones.

- **Hypothesis 3: Features cluster by importance.** This hypothesis explores whether features with similar predictive importance tend to group together within the feature space.

It is important to emphasize that these hypotheses are mutually independent. To test them, we propose two methodologies: (1) visualizing feature embeddings using UMAP and (2) clustering the features followed by dendrogram analysis to examine whether the resulting clusters are homogeneous (also called "pure") in terms of super-class, complexity, or importance.

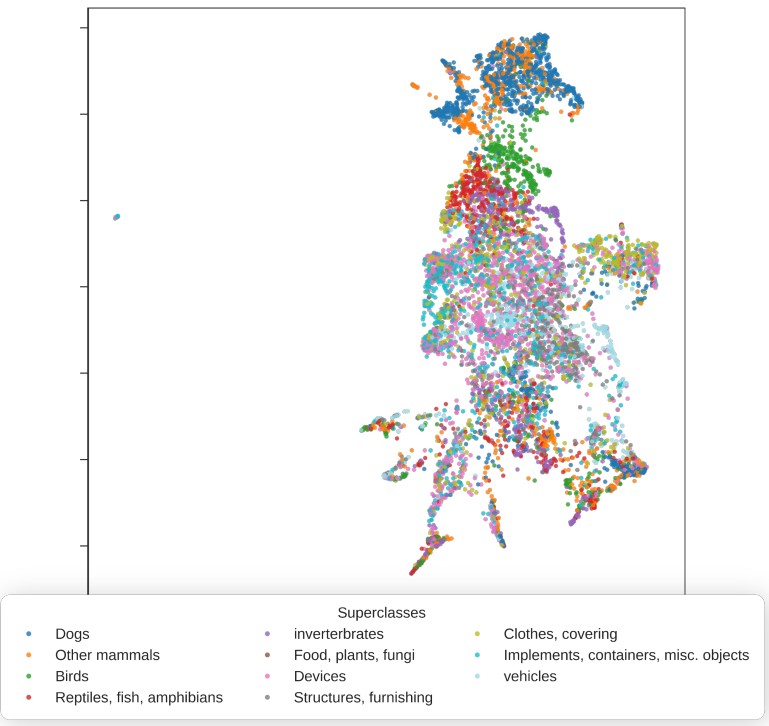

Figure 13: **Feature Similarity vs Super-Class.** Each point represents a concept, with its color indicating the associated super-class. Some super-classes such as birds, reptiles, dogs & other mammals form well-defined, tight clusters, suggesting that features belonging to them are close in the feature space. Others, such as device, clothes appear more dispersed. By comparing this figure with Figure 2, we can identify which meta-features are "pure" (belonging to a single super-class) and which are "impure" (spanning multiple super-classes). Interestingly, the "impurity" region seems to cover low-complexity and mid-complexity concepts such that Grass, Waves, Trees, Low-pixel quality detector which are not class-specific.

**Feature Similarity vs Super-Class.** Figure 13 illustrates the organization of features by their super-class. Each point is a feature, colored according to its super-class label. We observe that certain super-classes, like those associated with birds, dogs, reptiles form distinct and cohesive clusters, indicating a strong grouping within the feature space. Other region have features that encompassing a wider range of super-class, such that grass, waves, low-pixel quality detector. This reveals that some meta-features are predominantly associated with a single super-class ("pure"), while others span multiple super-classes ("impure"), reflecting the shared visual characteristics or multi-functional nature of those features.

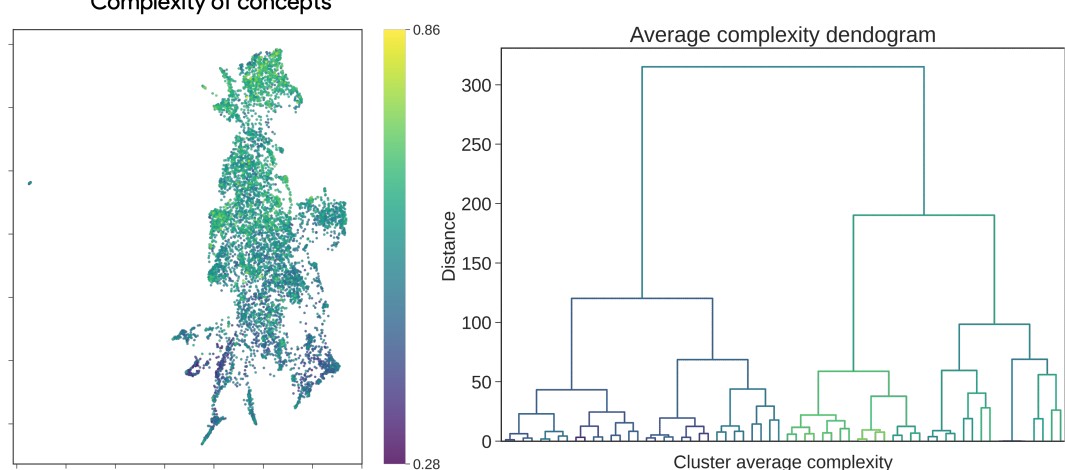

Figure 14: **Feature Similarity by Complexity. A)** Each point is a feature, colored by its complexity score. Distinct areas of the graph correspond to varying levels of complexity, suggesting a non-random distribution of feature complexity. For instance, animal-related features tend to have higher complexity, one could hypothesize that the fine-grained classification required for these categories are responsible for this complexity. **B)** A four-level dendrogram where each level further segments clusters and calculates the average complexity for each sub-cluster. A clear split by complexity appears at the first level and intensifies with depth, supporting the idea that some regions of the feature space are inherently more complex than others.

**Clustering by Complexity.** Figure 14 explores how features are organized based on their complexity. Panel **A** shows a UMAP visualization with points colored according to their complexity scores. The visualization reveals distinct regions of varying complexity, indicating that the distribution is structured. For instance, features related to animals display higher complexity, likely because these require fine-grained and precise detectors. In contrast, simpler features, such as color detectors or low-frequency patterns, cluster together in less complex regions. Panel **B** displays a dendrogram with four hierarchical levels, where each level introduces additional splits, and the mean complexity is calculated for each sub-cluster. The pronounced division in complexity at the first level, which sharpens as we delve deeper, suggests that the feature space is compartmentalized based on complexity. A promising direction for future research would be to align these complexity clusters with known visual cortical areas to explore potential correspondences.

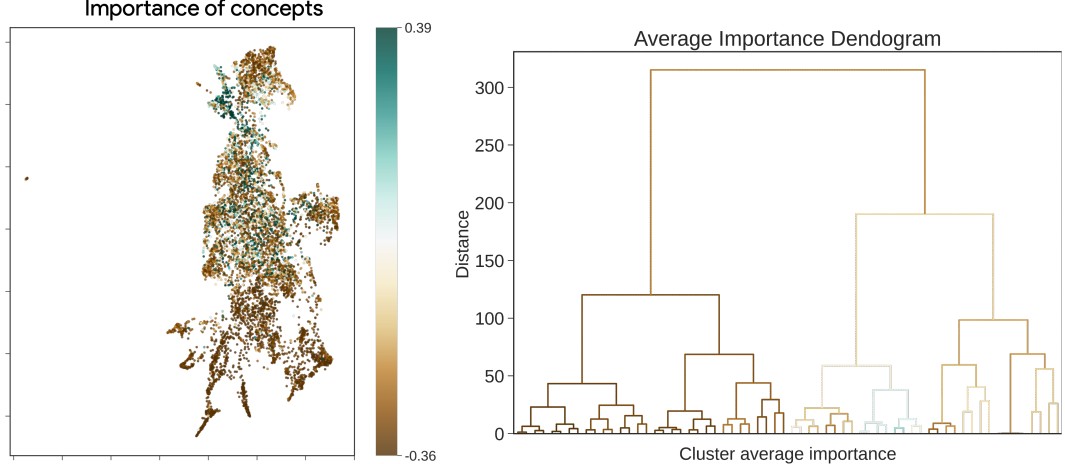

Figure 15: **Feature Similarity by Importance. A)** Each point represents a feature, with color indicating its importance. Distinct regions of the graph contain features of varying importance, particularly with more important features clustering at the top. **B)** A four-level dendrogram with sub-clusters evaluated by their average importance. We observe that from the first level, the dendrogram effectively splits features into groups of varying importance, corresponding to the upper and lower parts of the UMAP graph in Panel A.

**Clustering by Importance.** Finally, we investigate the third hypothesis: the presence of regions within the feature space that contain features with higher predictive importance. Figure 15 Panel A depicts a UMAP visualization where features are colored by their importance. The graph shows a clear structure, with highly important features grouping in specific regions (e.g., the upper part of the graph), while less important features are distributed in other regions. Panel B provides the corresponding dendrogram, revealing that even at the first level of the hierarchy, features segregate into clusters of varying importance. Notably, low-complexity "support" features—such as grass, waves, and low-pixel-quality detectors—tend to form cohesive clusters. Meanwhile, more predictive features, like animal-related features that drive classification, group together in another distinct region. This raises a crucial question: is this clustering merely correlated (i.e., based on shared visual aspects like context or background), or does it reflect a causal relationship in the model's predictive structure? This question remains an open avenue for future investigation.

# I  Local vs Distributed Encoding

Neural networks exhibit a diversity in how features are encoded, ranging from local to distributed representations [25, 29]. In the local encoding scenario, a single neuron is primarily responsible for encoding a feature. On the other hand, in a distributed encoding scheme, features are represented by the coordinated activity of multiple neurons, often deviating from canonical axis-aligned directions. More specifically, features could be distributed across many neurons—sometimes densely, or in a pseudo-distributed fashion—rather than being localized to a specific neuron. This theory is one of the main motivations behind employing overcomplete concept extraction methods [18, 32, 92, 35].

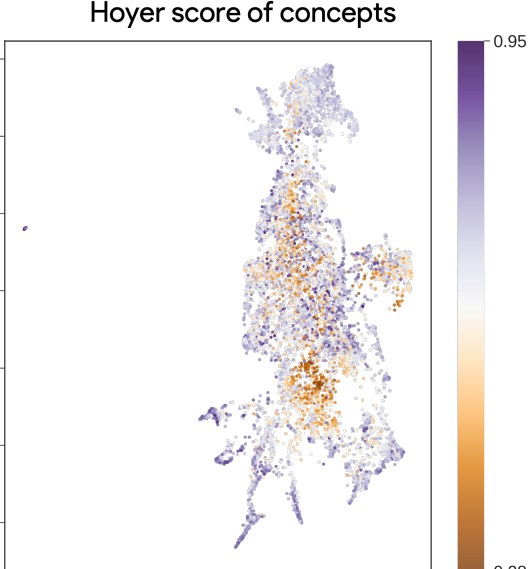

Figure 16: **Local vs Distributed Encoding.** Each point represents a feature, with color indicating its Hoyer score. Higher scores suggest a more "local" representation, where a feature is primarily encoded by a single neuron. Lower scores indicate a distributed representation across a population of neurons. Interestingly, some features have scores near 1, implying near-complete localization, while others are more distributed. This variation highlights the diversity in encoding across features.

The distinction between local and distributed encoding is critical, particularly when extracting overcomplete features from a large-scale dictionary. In our analysis of 10,000 features, we assess whether these features are encoded in a local or distributed manner. A local encoding would imply that the feature vector $d$ is aligned with a canonical vector, i.e., $d \in \{e_1, ..., e_n\}$, where $n = |\mathcal{A}_\ell|$ is the dimensionality of the activation space and $e_i$ represents the one-hot canonical vector for the $i$th neuron. In contrast, a fully distributed feature would be characterized by non-zero values across all neurons, with no alignment to any single axis.

To quantify the extent of this local or distributed encoding, we use the Hoyer score. This score, which ranges between 0 and 1, captures the sparsity of a vector by comparing its $\ell_1$-norm to its $\ell_2$-norm, with a correction term for normalization. Formally, the Hoyer score is given by:

$$\text{Hoyer}(\boldsymbol{d}) = \frac{\sqrt{n} - ||\boldsymbol{d}||_1/||\boldsymbol{d}||_2}{\sqrt{n} - 1}.$$

For each feature in our dictionary $\mathbf{D} = \{\boldsymbol{d}_1, ..., \boldsymbol{d}_{10,000}\}$, we compute the Hoyer score to determine whether the feature is locally encoded (score near 1) or distributed (score near 0).

Figure 16 illustrates the results of this analysis, revealing significant variability in the degree of distribution among features. Some regions of the feature space show highly distributed encoding, while others exhibit more local representations. This analysis was conducted on a ResNet50 model, and it is possible that the distribution of encoding strategies varies across different architectures.

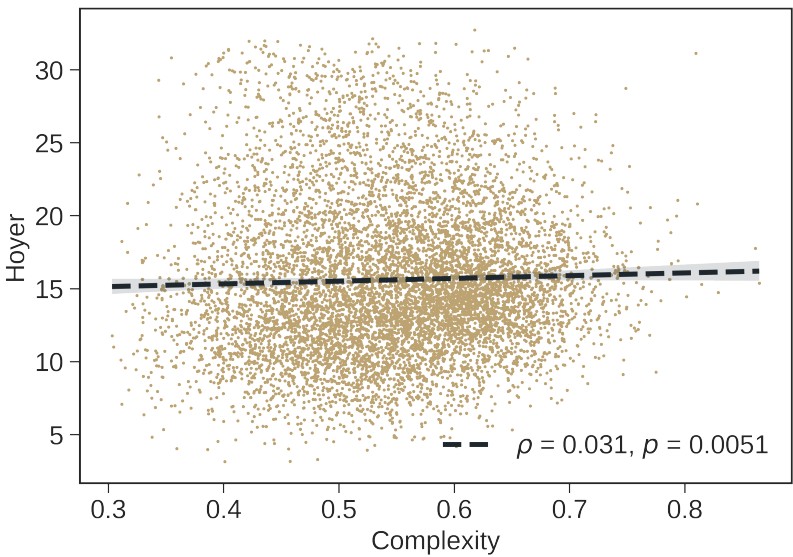

Figure 17: **Feature Complexity vs. Distributed Encoding.** We show that there is no clear relationship between feature complexity and the degree of distributed encoding. Whether a feature is encoded by a single neuron or distributed across multiple neurons does not seems to be determined by its complexity.

## J  Replicating the feature flow results with other measures and models.

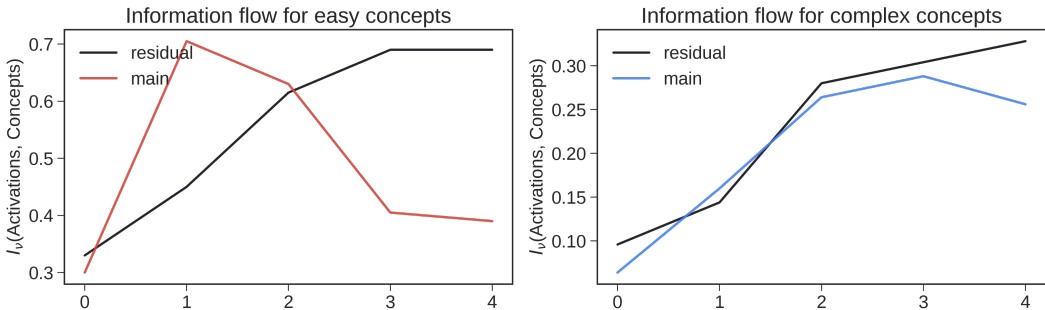

Figure 18: **Replication of Figure 4 Using $\mathcal{V}$-Information.** As described in Section 4, we replicate the analysis of feature flow and complexity using $\mathcal{V}$-information as a measure.

A legitimate consideration arises when revisiting Section 4, where we introduced the hypothesis that simpler features are primarily carried through the residual connections of a network, while more complex features are progressively constructed through interactions between the main branch and residual connections. The measure we originally used to support this hypothesis was CKA, which serves as a proxy to assess the similarity between the activations at a certain stage of the model and the final state of a concept. However, one might wonder why not use $\mathcal{V}$-information directly.

Figure 18 presents the results of this replication. Although the scale of the values differs from those in Section 4, the overall trend remains consistent. The left panel shows the dynamics of simpler features. Early in the network, the main branch carries a significant portion of the $\mathcal{V}$-information, which diminishes as the simpler features are gradually "passed along" through the residual connections. Notably, even as the information content decreases, the absolute quantity of $\mathcal{V}$-information remains higher for simple features compared to complex ones. This indicates that while simpler features are transported through the residual connections, they are not entirely depleted of their information content. The right panel of Figure 18 demonstrates the progressive construction of more complex features. Here, we see that both the main branch and residual connections contribute to the gradual accumulation of information necessary for these complex features. This supports our initial hypothesis: complex features do not traverse the network intact but are built up in a cumulative process, drawing on multiple layers and branches to form intricate representations.

For compute consideration, this replication was conducted using a different experimental setup than that of Section 4. For this analysis, we employed the validation set of ImageNet to build the dictionary instead of the train set. Both the dictionary and the model were different from those used in the main body of the paper. Specifically, the model used here was the ResNet50 implementation from the Keras library [26]. Despite these differences in experimental conditions, the overarching trends observed in our original CKA-based analysis are preserved, bolstering the validity of the flow hypothesis.

## K  Kolmogorov, Levin and $\mathcal{V}$-information

In this section we recall some of the most important complexity measures like Kolmogorov complexity, its computationally tractable counterpart the Levin complexity, and finally we underline the epistemic similarity between these concepts in deep learning.

**Kolmogorov complexity [55]**   is a measure of the complexity of an object. The objects (image, video, text, pdf, etc.) can be ecnoded as a sequence $(u_n) \in \Sigma^{\mathbb{N}}$ of symbols over a finite alphabet $\Sigma$. A program is a finite sequence $P \in L$ written in language $L \subset \Sigma^*$ (e.g a Python source file). Kolmogorov complexity $K_L^{(\infty)}(u_n)$ is the length of the shortest program $P : \mathbb{N} \to \Sigma^*$ that produces

---

There exists numerous variants of this definition with slightly different behaviors [65]. Since deriving theoretical results is not the focus of our work, we decided to tradeoff precision for simplicity of exposition.

the $n$-th first terms of the sequence $(u_n)$:

$$K_L^{(\infty)}(u_n) \overset{\text{def}}{=} \min_{P(n)=u_n} |P|. \tag{6}$$

Intuitively, if the sequence is highly compressible, the program will be short. For example, the sequences $[1, 2, 3, 4, \ldots]$ or $[2, 4, 8, 16, 32, \ldots]$ are few lines of code in most programming languages. Conversely, if the sequence is purely random, then no finite-length program exists. The digits of $\pi$, seemingly without structure, are not *Kolmogorov random* since there exist short programs computing them. The famous Cantor's diagonal argument [20] shows that most sequences are random, since no bijection exists between $\Sigma^*$ (countably infinite) and $\Sigma^\mathbb{N}$ (cardinality of the continuum). The definition implicitly assumes a specific computation model (Python interpreter, C++ compiler, Turing machine) to describe the language. However, by definition Turing-complete models can simulate each other, which implies there exist a universal constant $\mathcal{C}(\text{Python}|\text{C++})$ such that for all sequences $u_n$ we have $K_{\text{Python}}^{(\infty)}(u_n) \leq K_{\text{C++}}^{(\infty)}(u_n) + \mathcal{C}(\text{Python}|\text{C++})$. This constant corresponds to the length of a Python interpreter written in C++ for example. In general, this holds for any other pair of languages. Therefore, if $K_L^{(\infty)}(u_n) \to +\infty$ as $n \to \infty$ for some language $L$, then it is true in every other language: intrinsic randomness is *universal* in this sense [98].

**Levin complexity.**    Kolmogorov's complexity suffers from an important drawback: it is not Turing-computable. Put another way, there exists no algorithm that computes $K^{(\infty)}$. Fortunately, by *regularizing* $K^{(\infty)}$ appropriately it is possible to make it computable. Levin [64] proposed to regularize the cost with the runtime $T(P, n)$ of program $P$ on input $n$. This is the *Levin complexity*:

$$K_L^{(T)}(u_n) \overset{\text{def}}{=} \min_{P(n)=u_n} |P| + \log_{|\Sigma|} T(P, n). \tag{7}$$

This modification makes $K^{(T)}(u_n, L)$ computable with the **Levin Universal Search** algorithm (see Alg. 1). Informally, instead of looking for *a* shortest program, this algorithm seeks algorithms that run *fast* among those *who are shorts*. It is obtained by iterating over lengths $i \in \mathbb{N}$, and by running exactly one step of computation of all these programs in parallel. The first program $P$ that halts on $u_n$ minimizes $K^{(T)}$. This is a central property of Levin's universal search: *the first programs found are the simplest and the ones requiring the lesser compute* [4, 108, 13].

---

**Algorithm 1 : Levin Universal Search**

---

**Input**: sequence $(u_n) \in \Sigma^*$
**Output**: program $P$ minimizing $K_L^{(T)}$
1:  $S \leftarrow \varnothing$
2:  **for** $i \in \mathbb{N}$ **do**
3:      **for each** program $P \in (\Sigma^i \cap L)$ **do**
4:          $S \leftarrow S \cup \{P\}$
5:      **for each** $P \in S$ in parallel **do**
6:          Run $P$ for exactly 1 step.
7:          **if** $P$ halts on $u_n$ **then**
8:              **return** $P$

---

**Deep learning and simplicity bias.**    The links between algorithmic information theory and deep learning have been a recurring although spurious interest throughout the years [94, 95, 75, 61, 68, 40]. Neural networks are a special kind of program, composed of the source files required for inference, and the weights embedded in the network. Therefore, results related to the complexity of sequences apply transparently. Program length (Kolmogorov) and program runtime (Levin) are tightly linked since deeper and wider networks also consume more FLOPS during inference. Smaller networks implement simpler programs. Similarly, features that can be decoded "early" in the network are simpler than those requiring all the layers. The idea is often coined as Minimum Description Length (MDL) principle [16], Occam's Razzor, or even simplicity bias [49].

---

As for many things in machine learning, regularization helps.

# L  Limitations

**Task.**  Here, we have studied a specific CNN architecture, ResNet50. In future experiments, it will be useful to investigate whether other model families exhibit similar feature learning, including in domains beyond vision.

**Architecture.**  The residual connections of the ResNet are shared by other architectures like ConvNext [67] or Vision Transformers (ViT) [28, 14]. The works of [109] indicate that findings from convolutional models may transfer to ViTs. Furthermore, the work of [116] suggest that features in convolutional networks and ViT are of similar nature. However, [90] found significant quantitative differences between the layers of ResNets and ViTs, highlighting the need for further empirical testing.

**Training Dynamics**  The observed dynamics of feature learning, including the emergence of complex features and the reduction in the complexity of important features later in training, are based on a specific training schedule and set of hyperparameters. To accurately attribute these findings, a more comprehensive study is required to evaluate the role of various factors such as the learning rate scheduler and weight decay. Future research should systematically investigate how these and other training parameters influence feature complexity and importance.

**Nested predictive families.**  Our complexity metrics rely on the hypothesis that the different predictive families associated to the network up to depth $f_\ell(\mathbf{x})$ are nested, i.e. that stacking more layers strictly increases expressiveness. This is highlighted in the relevant assumption in section 2. If this hypothesis is violated, the true complexity of a feature may be overestimated in deeper layers. This is typically the case at the early stages of training.

**Dictionary of features.**  Regarding the building of the dictionary using NMF, a previous study [32] has shown that the specific dictionary learning method yielded a favorable tradeoff between several criterions such as sparsity, reconstruction error, or stability. Other dictionary learning methods (like sparse-PCA, K-Means or sparse auto-encoder) may yield a bank of concepts with different properties.

