# OpenReview forum: "Understanding Visual Feature Reliance through the Lens of Complexity"
_NeurIPS.cc/2024/Conference — NeurIPS 2024 poster_

### Official Review · Reviewer_d2Wb · 2024-07-12

**Soundness:** 4
**Presentation:** 4
**Contribution:** 3
**Rating:** 8
**Confidence:** 3

**Summary:**

This paper proposes a method to measure the complexity of features extracted by deep learning models. This method is based on $\mathcal{V}$-information, an extension of Shannon's mutual information that takes the computational capabilities of a decoder into account. The proposed measure of feature complexity is inversely related to the cumulative $\mathcal{V}$-information across a model's layers, the intuition being that features that become available at earlier layers will accumulate a larger $\mathcal{V}$-information, while features that only become available late will have a smaller $\mathcal{V}$-information.
Features are extracted from a standard, ImageNet-trained ResNet50, from which an overcomplete dictionary is learned. These features are then clustered, and the mean complexity score for each feature is computed. Feature clusters with low, intermediate and high complexity scores are visualized, revealing that low-complexity features tend to be related to uniform colors or low-frequency information, intermediate-complexity features to local shapes such as eyes and noses and high-complexity features to highly structured shapes, such as insect legs.
Using the CKA between the features in the dictionary and the activations at different layers within the main branch and residual branch of the network, the authors find that Easy features tend to emerge in early layers and then copied to later layers through the residual branch, while Complex features steadily increase throughout layers in both branches.
Analyzing the dynamics of different features' emergence throughout training, complex features are found to emerge later than simple ones. Interestingly, analyzing the interplay between features' complexity and their importance in influencing the network's outputs, the authors find that (a) more important features tend to be less complex, and (b) the complexity of important features is reduced throughout training, suggesting that the network might be "compressing" the more important features.

**Strengths:**

- The paper presents an elegant measure of feature complexity in deep neural networks, which takes into account the networks' computational expressiveness in different layers.
- The paper is extremely well written, with detailed methods and supplementary materials, exhaustive analyses and clear visualizations.
- The Related Works section is particularly exhaustive and thorough.
- It successfully combines several recently proposed interpretability tools.
- The analysis of the interplay between complexity and importance throughout learning was particularly interesting, suggesting a mechanism for the compression of important features across learning epochs in deep neural networks.

**Weaknesses:**

- The proposed measure is closely related to the accuracy with which a feature can be linearly decoded from the layers of a network. I believe the paper would benefit from an explicit discussion of what, exactly, is the additional information provided by the proposed method which cannot be gained from directly looking at the "raw" decoding accuracy.
- For simplicity, the authors restrict their analysis to a single ResNet model. While this is an understandable choice, I am not sure about what the implications are for models which do not include residual connections. The residual stream is found to be central in "teleporting" simple features to later layers. The authors should explicitly discuss what they predict the pattern of results would be for networks without residual connections. For example, would simple features be overall less relevant to the network's responses? Or would the network implicitly implement a residual-like stream? The limitations of using this architecture exclusively are briefly acknowledged in the Limitations section, but the specific role of residual connections is not discussed there.
- The proposed measure is based on a loose assumption that the decodability of features tends to increase across layers. However, it is possible that certain features are _only_ available in early layers (for example, certain low-level image features might be discarded). These features would have a low cumulative $\mathcal{V}$-information, and might thus be erroneously classified as high-complexity. The authors should explicitly discuss whether this is a concern at all, and for what reasons.
- The plot in Figure 5B shows that more important features tend to become less complex as training progresses. As the authors have access to the features which are subject to this process, and those which are not, it would be extremely interesting to visualize them, showing what exactly is happening - are the same features being extracted at earlier layers? Or are they actually changing, and starting to resemble simpler features such as colors or edges? The answer to this question would clarify more precisely what the proposed complexity measure is capturing. Is it possible that features we visually evaluate as "complex" can be extracted at early layers if they are important to the network's task? Or is layer depth always correlated with visual complexity?

**Questions:**

No questions, beyond the requests for clarification listed in the weaknesses section.

**Limitations:**

Limitations are discussed adequately. Please refer to the weaknesses section for comments about limitations which I believe were not sufficiently discussed.

---

> ### Author Rebuttal · Authors · 2024-08-07
>
> > "The proposed measure is closely related to the accuracy with which a feature can be linearly decoded from the layers of a network. I believe the paper would benefit from an explicit discussion of what, exactly, is the additional information provided by the proposed method which cannot be gained from directly looking at the "raw" decoding accuracy."
>
> We are not sure which accuracy you refer to.
>
> The classification accuracy is not smooth, nor continuous, in network predictions. This typically makes its interpretation more noisy. Indeed, for hard tasks, the accuracy may be near the one of a random predictor, while the loss itself might be significantly lower than the one of a random predictor. Furthermore, accuracy only makes sense for classification tasks over discrete sets.  But predicting a concept coefficient is not as simple as predicting if it is “present” or “absent”. Instead, it is a continuum where a concept can be more or less influential. In this case, we face a regression task, as explained in Appendix C. This smoothness allows better handling of inaccuracies in the concept extraction stage.
>
> If you refer to MSE, it is indeed closely related to $\nu$-information in the Gaussian posterior case (appendix C), but thanks to the “optional ignorance” property of $\nu$-information, we are certain that small MSE can be attributed to knowledge of the input, and not to simplicity of the task.
>
> > "I am not sure about what the implications are for models which do not include residual connections. The residual stream is found to be central in "teleporting" simple features to later layers. The authors should explicitly discuss what they predict the pattern of results would be for networks without residual connections. For example, would simple features be overall less relevant to the network's responses? Or would the network implicitly implement a residual-like stream? The limitations of using this architecture exclusively are briefly acknowledged in the Limitations section, but the specific role of residual connections is not discussed there."
>
> Thanks for this suggestion. We added a paragraph in the limitations section to discuss the significance of residual connections, and the questions you raise.
>
> > "The proposed measure is based on a loose assumption that the decodability of features tends to increase across layers. However, it is possible that certain features are only available in early layers (for example, certain low-level image features might be discarded). These features would have a low cumulative -information, and might thus be erroneously classified as high-complexity. The authors should explicitly discuss whether this is a concern at all, and for what reasons."
>
> This is a good question. To clarify, we do not have this problem because we only seek to decode features available at the last layer. We create the dictionary at the penultimate layer, ensuring each feature exists at the end. We then calculate their complexity, avoiding issues with features that don't exist at the penultimate layer. However, you highlight an interesting point: what about features present at layer 2 but not at layer 5 for example? This is outside the scope of our current study, but an interesting question for future work.
>
> > "The plot in Figure 5B shows that more important features tend to become less complex as training progresses. As the authors have access to the features which are subject to this process, and those which are not, it would be extremely interesting to visualize them, showing what exactly is happening - are the same features being extracted at earlier layers? Or are they actually changing, and starting to resemble simpler features such as colors or edges? The answer to this question would clarify more precisely what the proposed complexity measure is capturing. Is it possible that features we visually evaluate as "complex" can be extracted at early layers if they are important to the network's task? Or is layer depth always correlated with visual complexity?"
>
>  Concerning the tracking of type of features, we indeed have some figures available showing this, but the problem is that tracking concepts (i.e., not just averaging complexities but genuinely tracking and linking features across epochs) requires an algorithm with hyperparameters. Some features may disappear at certain points. We chose to show only the average and top and bottom complexity curves to avoid relying too much on the linking method for our analysis. However, this is an excellent remark, and we would like to investigate this further in future work.

---

> > ### Comment · Reviewer_d2Wb · 2024-08-13
> >
> > I thank the authors for their rebuttal. Their answers are very helpful to clarify the few points I was unsure about, or to acknowledge when open questions still remain (as in the role of residual connections and the tracking of features during training). Just for clarification, in my point about decoding accuracy I was indeed referring to MSE. I apologize for using the "accuracy" terminology which suggests a discrete classification, that was probably confusing.
> >
> > Given that my review was very positive in the first place, and my concerns were quite minor, I will keep the same score.

---

### Official Review · Reviewer_ozGz · 2024-07-12

**Soundness:** 4
**Presentation:** 2
**Contribution:** 3
**Rating:** 7
**Confidence:** 4

**Summary:**

The current work proposes a method to assess the complexity of features in a representation space in terms of usable information.  The measure of complexity is related to how far back in the layers of a trained network one can find information about the feature that is recoverable by a linear decoder.  Equipped with a measure of complexity per feature, the authors grab 10k distributed features in the penultimate layer of a ResNet50 and then examine various relationships such as the growth in complexity over the course of training, the relative importance of simple features in determining the output, and the “flow” of simple features through the residual backbone of the network.

**Strengths:**

The premise -- of a practical way to assess complexity via tiers of usable information, and then its employment as a route to a better understanding of learned features -- is very interesting.  To evaluate the complexity measure, the authors cleverly use the trained layers in the network to approximate a hierarchy of function classes, such that the amount of usable information between the input and the feature must increase as you move deeper into the network, and at each level all that is needed is a correlation measurement.  The complexity measure is thus reasonable and straightforward to measure in practice.  The authors are upfront about a central assumption related to optimal processing by the trained layers.  The writing is clear and the references to related literature are thorough and extensive.

**Weaknesses:**

Many of the analyses are questionable, in my opinion, making the work seem to have a strong premise with a lackluster follow up.  I will be happy to be rebutted on these points with justification and clarifications during the discussion period.

Specifically, the “what” and “where” analyses are odd to me.  My main gripes are below, with smaller points in the "questions" section.
- Regarding the qualitative exploration of dictionary vectors, and their grouping into clusters (for “what”): is there a good reason to think relative positions mean anything in feature space, and therefore that clustering is meaningful?  Were the dictionary vectors normalized beforehand, or else did the variable magnitude affect the UMAP and clustering?  Why were 150 clusters used?  Did a sweep over k or any other analysis indicate that the vectors are indeed clustered to some degree, and 150 is an appropriate choice?  Unless I missed it, it’s not explained how the meta-clusters were labeled -- was it just visual inspection, and the remaining 120 clusters simply weren’t interpretable?
- Regarding the use of CKA for “where”: The proposed complexity measure is already based on how far back in the processing the feature is still recognizable to a linear probe.  Why show CKA instead of the usable information as a function of depth for simple vs complex features?  It’s not intuitive to me to infer anything about the similarity of the “residual” branch aka simply the feature vec one layer deeper, compared to that of the delta (“main branch”), which is not used as a feature by the network.  It seems far more straightforward to look at any changes in the usable information as a function of depth as arising from the delta at that layer.  Why not just show the curves for each of the features?  The curves would also show the way in which usable information is growing with depth -- is it gradual, or stepped?  All said, it’s hard to find much insight from the results of Fig 4, and a straightforward alternative exists.

**Questions:**

- In Appendix A: “The dictionary D was designed to encapsulate 10 concepts per class”.  Does this mean the NMF was run separately for each class, and then the dictionaries were all combined?  If that’s the case, are there multiple copies of some features (e.g. “wheel” being found for every vehicle class)?
- How were the visualizations of Fig 1 produced, particularly the earlier manifestations z2 and z3?  How should we interpret a feature’s identity in an earlier layer when it is not recognizable in that earlier layer (and therefore a function of many features)?
- Fig 6: What are we supposed to infer from the epoch 1 results?  If everything is still essentially random at this stage of training, then we're just seeing meaningless values plotted against each other, right?
- The inhibitory/excitatory results (Fig 12) are striking -- **every single inhibitory feature has complexity less than ~0.5?**  While the other findings are rough trends extracted from point scatter, this finding is relegated to the appendices and given minimal attention?

Questions out of interest rather than needing to fill a gap in the paper:
- Regarding the higher importance of simpler features: are they simply present more often, or are they more influential?  I wonder if there's a frequency-complexity relationship that would be insightful.
- Did you do any of the same analyses with the individual neurons in the penultimate layer (i.e. comparing local vs distributed)?

**Limitations:**

There is a reasonable discussion of limitations in the Appendices.

---

> ### Author Rebuttal · Authors · 2024-08-07
>
> We greatly appreciate the time and effort you invested in reviewing our work; thank you.
>
> > "Regarding the qualitative exploration of dictionary vectors, and their grouping into clusters (for “what”): is there a good reason to think relative positions mean anything in feature space, and therefore that clustering is meaningful? Were the dictionary vectors normalized beforehand, or else did the variable magnitude affect the UMAP and clustering? Why were 150 clusters used? Did a sweep over k or any other analysis indicate that the vectors are indeed clustered to some degree ? "
>
> Clustering was used to qualitatively inspect what complex and simple features look like, and it represents a qualitative part of our study. Clustering in embedding space is a common practice, and normalization does not change the cluster membership. We found that meta-clusters were robust to choice of cosine or L2 normalization.
>
> Regarding the choice of 150 clusters, we aimed to avoid overcrowding figures and selected 30 meta-features that span different degrees of complexity, sampling randomly at each level. Each cluster was qualitatively interpreted, but the labeling of clusters is subjective. We didn't use external models for labeling, and while these labels provide useful insights, they are inherently qualitative and should be interpreted as such. We have clarified this aspect in the manuscript to emphasize the qualitative nature of this part of the analysis and to acknowledge the limitations of the clustering and labeling process.
>
> > "Regarding the use of CKA for “where”: The proposed complexity measure is already based on how far back in the processing the feature is still recognizable to a linear probe. Why show CKA instead of the usable information as a function of depth for simple vs complex features? It’s not intuitive to me to infer anything about the similarity of the “residual” branch aka simply the feature vec one layer deeper, compared to that of the delta (“main branch”), which is not used as a feature by the network."
>
> Good question as you are indeed correct. We acknowledge that $\nu$-information could provide a valuable perspective, and could be used for that. However, to avoid circular reasoning and to employ a well-established metric in the literature, we chose CKA.
> Moreover, our results with $\nu$-information are consistent with those obtained using CKA, and we have included these additional findings in the appendix as suggested. The choice of CKA was also driven by its established use in analyzing neural network activations, ensuring our findings are interpretable and comparable within the broader research context.
>
> > "In Appendix A: “The dictionary D was designed to encapsulate 10 concepts per class”. Does this mean the NMF was run separately for each class, and then the dictionaries were all combined? If that’s the case, are there multiple copies of some features (e.g. “wheel” being found for every vehicle class)?"
>
> The dictionary design process utilized the published Craft method, ensuring 10 concepts per class to achieve balanced reconstruction across classes. While this approach can result in duplicated concepts, it ensures comprehensive coverage of class-specific features. We believe this redundancy does not detract from the overall utility of the dictionary; rather, it enhances the robustness of feature representation by capturing subtle variations within shared concepts across different classes.
>
> > "Fig 6: What are we supposed to infer from the epoch 1 results? If everything is still essentially random at this stage of training, then we're just seeing meaningless values plotted against each other, right?"
>
> In this figure, we contrast features at the beginning versus end of training, to get a picture of what has shifted. The end of epoch 1 represents the point where the dataset has been seen once, and several gradient steps have been taken. The features used at the end of epoch 1 have equal chances of being simple or complex (decodable at layer 1 or layer 10, for example). In contrast, by the end of training, we observe a simplicity bias, where important features are more likely to be decodable early. This demonstrates how the network evolves to prioritize simpler features over time, highlighting the dynamic nature of feature learning and the emergence of important features throughout the training process.
>
> > "The inhibitory/excitatory results (Fig 12) are striking -- every single inhibitory feature has complexity less than ~0.5?"
>
> Thank you for this remark. We also find this observation intriguing. To be honest, we are not certain whether this is a specific property of ResNet50, ImageNet (because of 1k classes), or a more general phenomenon. This warrants further investigation and could be a fruitful avenue for future research.
>
> > "Regarding the higher importance of simpler features: are they simply present more often, or are they more influential? I wonder if there's a frequency-complexity relationship that would be insightful."
>
> This is an excellent question. Previous work has already pointed to these issues [1], and concurrent research has posed similar questions with controlled toy examples [2]. It appears that the relationship between frequency and complexity is indeed complex, involving both factors. Simpler features may be more frequently present and also more influential in driving the network's decisions. We believe this is a fundamental question for deep learning.
>
> [1] Hermann, K., Mobahi, H., Fel, T. Mozer, M. (2023). On the Foundations of Shortcut Learning.
> [2] Lampinen, A. K., Chan, S. C., and Hermann, K. (2024). Learned feature representations are biased by complexity, learning order, position, and more.

---

> > ### Comment · Reviewer_ozGz · 2024-08-09
> >
> > Thank you for the responses.  I would have liked to see in a rebuttal pdf the usable information vs depth comparison to CKA ("we have included these additional findings in the appendix as suggested"), but I'll take your word for it, and look forward to checking it out in the published version.
> >
> > All told, this paper (refreshingly) gave me a lot to think about, and I'm more convinced after the response about the soundness of the analyses.  I'll raise my score to a 7; good luck with acceptance.

---

> ### Author Response · Authors · 2024-08-12
>
> Thank you for your feedback! We're glad our responses helped clarify the analysis. We'll make sure the comparison with CKA is included in the appendix as suggested. We appreciate your support and the score increase. Thanks again, and we hope the final version meets your expectations!

---

### Official Review · Reviewer_mfVA · 2024-07-12

**Soundness:** 3
**Presentation:** 2
**Contribution:** 2
**Rating:** 6
**Confidence:** 1

**Summary:**

The paper introduces a metric based on V-information to measure feature complexity in deep learning models. Using ResNet50 trained on ImageNet,  it explores feature spectrum, training dynamics, network flow, and decision impact.

The study also highlights the role of simplicity bias and the evolution of feature importance in neural networks.

**Strengths:**

1) Clear and well defined objective
2) Informative Figures
3) Comprehensive supplementary material

**Weaknesses:**

1) Not Easy To Follow On Math: The paper's mathematical sections are difficult to follow. Clearer and intuitive explanations would improve accessibility.
2) Limited Model Diversity: The study focuses solely on ResNet50, ignoring modern architectures like depth-wise separable CNNs.

**Questions:**

1) The paper states that both $z$ and $D^*$ are positive due to the use of Non-Negative Matrix Factorization, which aligns with the nature of ReLUs. Could you elaborate on why negative values for $D^*$ wouldn't work, despite the positive nature of ReLU? It's not clear why allowing $z$ and $D^*$ to be negative would be problematic, especially considering that $(-z)(-D^*) = zD^*$.

2) In line 126, $z$ is defined as $z \in \mathbb{R}$. Given that $z$ is described as positive elsewhere, wouldn't $z \in \mathbb{R^+}$ be more precise? Or are you intentionally allowing negative values for $z$?

3) The nature of $z_1$, $z_2$, and $z_3$ in Figure 1 is unclear. If these represent the $i$-th element of $z$, they would be scalars, which doesn't seem to align with what's depicted. Could you clarify what these variables represent?

4) Figure 1 shows feature visualization across different layers, and the paper mentions earlier layers elsewhere. However, line 622 states that feature extraction was done only for the penultimate layer. Could you explain this and clarify how earlier layer information was obtained if extraction was limited to the penultimate layer?

**Limitations:**

The authors have addressed limitations in the appendix.

---

> ### Author Rebuttal · Authors · 2024-08-07
>
> > "Not Easy To Follow On Math: The paper's mathematical sections are difficult to follow. Clearer and intuitive explanations would improve accessibility."
>
> We thank the reviewer for this feedback. We strove to keep the math portions of the paper as accessible as possible, but welcome feedback if there were particular points of confusion.
>
> > "The paper states that both and are positive due to the use of Non-Negative Matrix Factorization, which aligns with the nature of ReLUs. Could you elaborate on why negative values for wouldn't work, despite the positive nature of ReLU? It's not clear why allowing and to be negative would be problematic, especially considering that."
>
> We are applying a published method widely recognized to generate concepts that are interpretable, sparse, and compositional; see [1, 2] for an overview of the large literature on the subject.
> Theoretically speaking, the importance of using a consistent semi-ring on both sides (non-negative semi-ring or negative semi-ring, as you pointed out) ensures compositionality and interpretability, preventing the cancellation of concepts (we only add value on top of each other; they don't cancel).
> Practically speaking, NMF (i) aligns well with the properties of ReLU activations, which are sparse and positive. NMF (ii) allows us to decompose activations without orthogonality constraints, which is crucial since model activations can collapse, making concepts non-orthogonal. Moreover, NMF (iii) supports compositionality, enabling multiple concepts to coexist within an activation.
>
> [1] Lee, D. D., & Seung, H. S. (1999). Learning the parts of objects by non-negative matrix factorization. Nature.
> [2] Gillis, N. (2014). Nonnegative Matrix Factorization.
>
> > "In line 126, is defined as. Given that is described as positive elsewhere, wouldn't be more precise? Or are you intentionally allowing negative values for?"
>
> This is a good remark. We have corrected this inconsistency to ensure clarity.
>
> > "The nature of, , and in Figure 1 is unclear. If these represent the -th element of, they would be scalars, which doesn't seem to align with what's depicted. Could you clarify what these variables represent?"
>
> You are correct, and we apologize for the lack of clarity. $z_1, z_2$, and $z_3$ ​ are indeed scalar values representing directions in the activation space (also called “concepts”). The images shown are feature visualizations, which are optimized to maximize the activation of these concepts (i.e., finding $x$ such that $x = \arg\max z_i(x)$. We have clarified this in the manuscript to avoid confusion.
>
> > "Figure 1 shows feature visualization across different layers, and the paper mentions earlier layers elsewhere. However, line 622 states that feature extraction was done only for the penultimate layer. Could you explain this and clarify how earlier layer information was obtained if extraction was limited to the penultimate layer?"
>
> Thank you for this remark. We have corrected this inconsistency in the manuscript. Feature extraction was indeed performed at the penultimate layer. The visualizations in Figure 1 aim to decode features $z_1, z_2$, and $z_3$ that exist at the penultimate layer at lower layers, such as blocks 1 and 3 of the ResNet50. We use linear probing at these earlier layers to visualize how features develop and are transformed across the network. This method clarification has been added to the manuscript.

---

> > ### Comment · Reviewer_mfVA · 2024-08-12
> >
> > I thank the authors for their rebuttal and their time. Their clarification were indeed insightful for better understanding the paper.
> >
> > After reading other discussions, and reading the paper one more time, I'm even more unsure of my assessment. I will stand by my current rating as I think the authors put a lot of effort in this paper, and their results are reasonable, but I will downgrade my confidence to 1.
> >
> > I hope the best for the authors.
> >
> > Best regards,

---

### Official Review · Reviewer_aiER · 2024-07-12

**Soundness:** 2
**Presentation:** 3
**Contribution:** 2
**Rating:** 4
**Confidence:** 3

**Summary:**

The paper introduces a novel metric for quantifying feature complexity in deep learning models, specifically focusing on an ImageNet-trained ResNet50 model. This V-information-based metric captures whether a feature requires complex computational transformations for extraction.  The study addresses four key questions:
(1) The appearance of features as a function of complexity.
(2) The learning timeline of these features during training.
(3) The flow of simple and complex features within the network.
(4) The relationship between feature complexity and their importance in the model's decision-making.
The study reveals that simpler features dominate early in training and are transported through the network via residual connections, while more complex features emerge gradually and require more computational effort. Interestingly, the most important features tend to be simpler and become accessible earlier in the network, suggesting a sedimentation process.

**Strengths:**

(1) Originality: The introduction of the V-information metric for assessing feature complexity is novel and provides a fresh perspective on understanding neural network behavior. The exploration of feature complexity across training epochs and network layers adds significant depth to existing knowledge.

(2) Clarity: The paper is well-organized, clearly presenting its goals, methods, and findings. Visualizations and qualitative analyses effectively illustrate the differences between simple and complex features.

(3) Significance: Understanding feature complexity and its impact on model performance and decision-making is crucial for developing more interpretable and efficient models. This work contributes to explainable AI by providing insights into how features are learned and utilized.

**Weaknesses:**

(1) Generalizability: While the findings are insightful, they are based on a single architecture (ResNet50) and dataset (ImageNet). The results might not generalize to other models or tasks without further validation.

(2) Complexity Metric: The assumption that each layer optimally represents features for downstream linear probes may not hold universally, potentially leading to overestimated complexity scores in some cases. More empirical validation across different models and tasks could strengthen the proposed metric's robustness.

(3) Temporal Analysis: The focus on specific epochs (e.g., epoch 90) may overlook important dynamics occurring at intermediate stages of training. A more granular analysis could provide a clearer picture of feature evolution, i.e., Can you analyze the changes in importance during the training process？

**Questions:**

(1) How would the proposed V-information complexity metric perform on different neural network architectures, such as Transformers or GANs?
(2) Can the authors provide more details on how the complexity metric correlates with other existing complexity measures in literature? Which one provides better insights?
(3) How does the model's simplicity bias, as observed in the study, impact its generalization performance on unseen data?
(4) Are there specific strategies that could be employed to mitigate the simplicity bias and encourage the learning of more complex, yet important features?
(5) Does the conclusion of section 4 still hold on networks outside ResNet? How to explain the architecture of the transformer regarding the residual connection?
(6) Why does the horizontal axis in Figure 4 show several blocks instead of all blocks?

**Limitations:**

While the paper provides significant insights into feature complexity in deep learning models, there are several limitations that need to be addressed:

Model and Dataset Scope: The study focuses solely on a ResNet50 model trained on the ImageNet dataset. This raises concerns about the generalizability of the findings to other neural network architectures (e.g., Transformers, GANs) and different types of datasets. Future work should validate the proposed metric across a variety of models and tasks to ensure broader applicability.

Assumption on Optimal Representation: The complexity metric relies on the assumption that each layer in the network provides an optimal representation for downstream linear probes. However, this assumption may not always hold true, potentially leading to overestimated complexity scores. Further empirical validation and adjustments to the metric may be necessary to account for cases where this assumption is violated.

---

> ### Author Rebuttal · Authors · 2024-08-07
>
> > "While the findings are insightful, they are based on a single architecture (ResNet50) and dataset (ImageNet). The results might not generalize to other models or tasks without further validation." "The assumption that each layer optimally represents features for downstream linear probes may not hold universally, potentially leading to overestimated complexity scores in some cases. More empirical validation across different models and tasks could strengthen the proposed metric's robustness."
>
> This is a valid point, which we discuss in lines 325-331 and in Appendix I. We agree and would like to extend our study to include diverse models in the future. However, our perspective with this work is that understanding why ResNet50 generalizes on ImageNet is already a significant challenge.
>
> > "The focus on specific epochs (e.g., epoch 90) may overlook important dynamics occurring at intermediate stages of training. A more granular analysis could provide a clearer picture of feature evolution, i.e., Can you analyze the changes in importance during the training process?"
>
> Thank you for your comment. However, there might be a misunderstanding. Sections 5 and 6 of our paper directly address this issue. We indeed perform a dynamical study of features across epochs, and we show (i) in Section 5 that the most complex features emerge later in terms of epochs, while section 6 demonstrates both (ii) the simplicity bias over epochs and (iii) the dynamic compression of important features throughout the epochs.
>
> > "How would the proposed V-information complexity metric perform on different neural network architectures, such as Transformers or GANs?"
>
> We're not entirely sure if we understand the question. What do you mean by 'perform'? The metric, as introduced in the seminal paper [1], measures mutual information under a complexity constraint. We use it to determine the simplicity or complexity of features. How would you suggest evaluating the performance of our metric? If you're asking about applicability, our metric can be easily applied to transformers or GANs without any issues, as it only requires having access to intermediate activations.
>
> [1] A Theory of Usable Information under Computational Constraints, Xu et al., 2020
>
> > "Can the authors provide more details on how the complexity metric correlates with other existing complexity measures in the literature? Which one provides better insights?"
>
> You raise a valid point, as complexity can be defined in various ways. Recent research employs category theory to introduce a redundancy-based metric, which merges neurons until a distance gap surpasses a threshold, using this gap as a hyperparameter [1]. In Appendix E, we demonstrate a correlation between our metric and the general redundancy score presented in [2]. However, this method has two limitations: (i) it requires a hyperparameter, and (ii) features considered redundant by this metric may still be complex according to our measure. Our complexity metric is more focused on the computational operations needed to decode a feature optimally. In this regard, our complexity metric can be viewed as a relaxation of computational complexity metrics, such as Kolmogorov complexity.
>
> [1] Going beyond neural network feature similarity: The network feature complexity and its interpretation using category theory. Chen et al. (2023).
>
> [2] Diffused redundancy in pre-trained representations. Nanda et al. (2024)
>
> > "How does the model's simplicity bias, as observed in the study, impact its generalization performance on unseen data?"
>
> This phenomenon has been extensively studied in previous research, which tends to show that a simplicity bias can negatively impact model performance on unseen data. Models that overly rely on simpler features may fail to capture the more complex patterns necessary for robust generalization. We have incorporated references to these relevant studies [1,2] in our Related Work section to provide additional context and support for this observation.
>
> [1] The Pitfalls of Simplicity Bias in Neural Networks. Shah, H., Tamuly, K., Raghunathan, A., Jain, P., Netrapalli, P.
>
> [2] Overcoming Simplicity Bias in Deep Networks using a Feature Sieve. Tiwari R. ,Shenoy, P. (2023).
>
> > "Are there specific strategies that could be employed to mitigate the simplicity bias and encourage the learning of more complex, yet important, features?"
>
> One potential strategy we could imagine to mitigate simplicity bias using our metric is to backpropagate through the score of the $\nu$-information. This score is differentiable, allowing for backpropagation (backprop through single matrix inversion). While this method can theoretically prioritize the learning of more complex features, it presents significant challenges in terms of memory and scalability. We could imagine addressing this with an online score (or batch estimation) instead of a real Complexity score.

---

### Official Review · Reviewer_5SRh · 2024-07-13

**Soundness:** 2
**Presentation:** 3
**Contribution:** 1
**Rating:** 4
**Confidence:** 4

**Summary:**

This paper proposes a new measure of feature complexity based on an information-theoretic metric, the $\nu$-information metric. Utilizing this complexity measure, the paper shows (1) visualization of features of different complexities, (2) simple features are propagated through the residual connections to reach the final layer, (3) simple features are learned earlier than complex features during training, and (4) simple features usually have a higher importance (weight) to the output score of the model.

**Strengths:**

1.	The paper is well presented and is quite easy to follow. I appreciate the summary of different experiments into the words “what” “where” and ”when.”

2.	The paper provides abundant visualization to help readers grasp the intuitive behind simple features and complex features.

**Weaknesses:**

My major concerns for this paper are its coherence and significance.

1.	The paper lacks coherence because multiple components (e.g., metrics, algorithms) introduced in this paper do not come from the same theoretical framework. For example, the notion of feature complexity in this paper (the $\nu$-information metric) is drawn from information theory, while the method for extracting features (the Craft method) is based on non-negative matrix factorization and has little connection with information theory. The same issue applies to the feature visualization method for demonstrating features of different complexities, and CKA metric, and the importance metric $\Gamma(z_i)$. These metrics/algorithms are borrowed from different theoretical frameworks and contexts, so it is in doubt whether they can be used together. I would appreciate it if the authors are able to re-organize all components of the paper under a coherent framework, e.g., the information theoretic perspective (if at all possible).

2.	My second concern is about the paper’s significance, since many claims have been discussed in previous works and appears non-novel. The conclusion that simple features are usually color/edge detectors that are located in earlier layers is not surprising, nor do the conjecture that simple features tend to propagate through the residual connection. For the conclusion that simple features are learned earlier than complex features, there are many studies leading to this conclusion, from the perspective of spectral bias [cite1], game-theoretic interactions [cite2], or frequency [cite3]. Overall speaking, I do not learn new insights from the paper. Nevertheless, as mentioned in the 1st point in weaknesses, if the authors are able to re-organize the paper from the purely information-theoretic perspective, it will be a more intriguing work.

3.	Using K-means clustering to aggregate features into meta-features may lead to incorrect clustering results. K-means clustering is known for its poor performance for data clusters that are not circular-shaped and are of discrepant sizes. I am not sure how features are distributed in this paper’s experiments, but more advanced clustering strategies such as the spectral clustering or hierarchical clustering can be applied to mitigate this issue.

[cite1] Rahaman et al. On the Spectral Bias of Neural Networks. ICML, 2019.

[cite2] Liu et al. Towards the Difficulty for a Deep Neural Network to Learn Concepts of Different Complexities. NeurIPS, 2023.

[cite3] Xu et al. Frequency Principle: Fourier Analysis Sheds Light on Deep Neural Networks. ICLR, 2020.

**Questions:**

1.	The paper claims that simple features tend to propagate through the residual connection, while complex features tend to propagate through the main branch of the network. Then a natural question arises: how are simple and complex features propagate in CNNs without residual connections (e.g., AlexNet, VGG)?

2.	About the visualization of simple and complex features in Appendix B. The feature *fences* is among the most simple features, while the features *whiskers* and *dotted texture* are among the most complex features. However, I do not see an essential difference between the features *fences*, *whiskers*, and *dotted texture* from the sample images in Figure 7 and Figure 8. For example, *fences* and *dotted texture* both seem to consist of lines that cross over each other to form holes. Similarly, the *whiskers* feature is also composed of lines, although the lines do not cross over each other but appear parallel. Why is *fences* a simple feature, but *whiskers* and *dotted texture* complex features?

Another question is that previous studies show that CNNs are usually biased towards textures, i.e., they tend to learn textures as a shortcut solution, but why is the *dotted texture* here measured to be a complex feature?

---

> ### Author Rebuttal · Authors · 2024-08-07
>
> > "The paper lacks coherence because multiple components (e.g., metrics, algorithms) introduced in this paper do not come from the same theoretical framework. [...] These metrics/algorithms are borrowed from different theoretical frameworks and contexts, so it is in doubt whether they can be used together. I would appreciate it if the authors are able to re-organize all components of the paper under a coherent framework, e.g., the information theoretic perspective (if at all possible)."
>
> Regarding the different theoretical frameworks for the metrics and algorithms introduced, we believe that (1) these components can indeed be integrated together as is usually done and (2) we could if we wanted to write the entire framework from an information theoretic perspective. For instance, dictionary learning methods have a strong connection with information theory [1,2]. Additionally, CKA is based on HSIC, which is closely related to mutual information ($HSIC = MMD(P, \prod P_i$) and mutual information $KL(P, \prod P_i))$.  The framework of $\nu$-Informations is not tied-up to information theory alone, quite to the contrary. It inherits most of the theoretical properties of Shannon’s entropy (e.g …) but the possibility to specify any function class for the predictive family is what makes it powerful and applicable to a broad range of domains (including deep learning, but not only).
>
> It would be possible to reorganize the components under a unified framework, even if it is not the traditional approach for each component. Thank you for your suggestion; we are adding these insights in the appendix.
>
> [1]: B. Dumitrescu and P. Irofti, Dictionary Learning Algorithms and Applications
>
> [2]: A Personal Introduction to Theoretical Dictionary Learning, Karin Schnass
>
> > "My second concern is about the paper’s significance, since many claims have been discussed in previous works and appears non-novel. The conclusion that simple features are usually color/edge detectors that are located in earlier layers is not surprising, nor do the conjecture that simple features tend to propagate through the residual connection [...]Overall speaking, I do not learn new insights from the paper."
>
> You are correct that Section 3 qualitatively presents a range of simple, medium, and complex features. While some of these are new, some align with features we have learned about from the prior literature. Still, it is worthwhile to see that our method discovers them. Regarding Section 4, we do not claim novelty for the late appearance phenomenon, although this is the first explanation from a complexity perspective. As another reviewer noted, this provides a fresh viewpoint on the phenomenon. We believe that Section 6 presents two new contributions: the emergence of simplicity bias using $\nu$-Information during training and the observation that neural networks generalize by compressing important features.
>
> > "Using K-means clustering to aggregate features into meta-features may lead to incorrect clustering results. K-means clustering is known for its poor performance for data clusters that are not circular-shaped and are of discrepant sizes. I am not sure how features are distributed in this paper’s experiments, but more advanced clustering strategies such as the spectral clustering or hierarchical clustering can be applied to mitigate this issue."
>
> Thank you for your suggestion. We chose to use K-means based on the rationale that, in the collapsed space of a neural network, the choice of clustering algorithm has minimal impact on the results: studies have shown that feature distributions in this space are generally well-suited for K-means clustering [1].
>
> [1] "Neural Collapse: A Phenomenon in the Terminal Phase of Deep Learning" by Papyan et al., 2020
>
> > "The paper claims that simple features tend to propagate through the residual connection, while complex features tend to propagate through the main branch of the network. Then a natural question arises: how are simple and complex features propagate in CNNs without residual connections (e.g., AlexNet, VGG)?"
>
> This is indeed a relevant and intriguing question. Investigating whether identity functions or orthogonal transformations occur channel-wise in models like AlexNet and VGG could provide valuable insights. This topic warrants further exploration in future research, and has been noted in future directions.
>
> > "The feature fences is among the most simple features, while the features whiskers and dotted texture are among the most complex features. However, I do not see an essential difference between the features fences, whiskers, and dotted texture from the sample images in Figure 7 and Figure 8. For example, fences and dotted texture both seem to consist of lines that cross over each other to form holes. Similarly, the whiskers feature is also composed of lines, although the lines do not cross over each other but appear parallel. Why is fences a simple feature, but whiskers and dotted texture complex features?"
>
> We argue that fences represent repetitive local patterns, in contrast to whiskers or insect legs, which are unique, finely structured motifs. This distinction ties into the texture versus shape debate: texture pertains to repetitive patterns, while shape involves unique, widely distributed elements across an image, necessitating the extraction of more structured motifs.
>
> > "Another question is that previous studies show that CNNs are usually biased towards textures, i.e., they tend to learn textures as a shortcut solution, but why is the dotted texture here measured to be a complex feature?"
>
> Firstly, the dotted pattern falls within the medium complexity category, ranking 14th out of 30 meta-features. Although not highly complex, the appendix reveals that images responding to this pattern exhibit different orientations. This indicates that the pattern detects texture while also capturing it from multiple angles and viewpoints.

---

> > ### Comment · Reviewer_5SRh · 2024-08-13
> >
> > I would like to thank the authors for their response. The explanation for using the K-means method, and the complexity of the *fences* and *dotted texture* feature seem reasonable. Considering this effort, I have raised my score to 4.
> >
> > However, I cannot give a higher score because my concern for the coherence and significance of the paper remains. To further clarify the coherence issue, the main point of "multiple components introduced in this paper do not come from the same theoretical framework" is to justify that all these algorithms and metrics can used together and do not conflict with each other. For example, why choose the Craft method to extract features but not other methods? Is Craft truly compatible with the proposed $\nu$-information metric? The same question can be asked for other components: the feature visualization method, the CKA metric, the importance metric $\Gamma(z_i)$, etc. These components are expected to be organized in a coherent framework, or at least be justified for the particular choices, rather than appear like a combination of engineering techniques. I hope this paper can be more impactful in doing so.

---

> ### Author Response · Authors · 2024-08-13
> **Thank you for increasing your score**
>
> >  the main point [...] is to justify that all these algorithms and metrics can used together and do not conflict with each other
> > Is Craft truly compatible with the proposed v-information metric?
>
> Can you clarify what you mean by "conflicting with each other" ? None of these frameworks operate with mutually exclusive hypotheses.
>
> > These components are expected to be organized in a coherent framework
>
> This is rather ambitious, as it would require a comprehensive understanding of all phenomena arising in deep learning training. We believe our work provides a first step into the direction of identifying promising components of a "coherent framework".
>
> > why choose the Craft method to extract features but not other methods?
> > The same question can be asked for other components: the feature visualization method, the CKA metric, the importance metric , etc
>
> All these methods are published, and have demonstrated their effectiveness in deep learning benchmarks, including human experiments.
>
> We observe consistent results using different tools and theories. This diversity is a strength, not a weakness: this is precisely what makes us confident in the robustness of our observations. As all results and metrics point toward the same direction, we are confident to not overfit a single method.

---

### Author Rebuttal · Authors · 2024-08-07

### General comments

Thank you to the reviewers for taking the time to read and review our paper. Your critiques are sharp and insightful. You found the paper "extremely well-written" and "well presented and easy to follow." Regarding the results, you noted that it "adds significant depth to existing knowledge," supported by "thorough and extensive references to related literature." Finally, you found the results on the compression of important features "particularly interesting, suggesting a mechanism for the compression of important features across learning epochs in deep neural networks.".

However, we acknowledge that you have raised certain points and identified weaknesses in the paper. During the rebuttal period, we carefully examined the critiques provided by all five reviewers. We have diligently incorporated the necessary citations and clarifications into our Related Work and Discussion sections. We believe we have addressed all the comments in a satisfactory manner

We will now proceed to address each reviewer's comments directly, providing detailed justifications and clarifications for the points raised. We appreciate the opportunity to improve our paper through this constructive feedback.

### About dataset and model

> "Does the conclusion of section 4 still hold on networks outside ResNet? How to explain the architecture of the transformer regarding the residual connection?"
> "Limited Model Diversity: The study focuses solely on ResNet50, ignoring modern architectures like depth-wise separable CNNs."

We recognize this concern, as outlined in our initial two limitations points. Analyzing ResNet50 during training, as done here, already entails examining over a million features from a standard model used in practice. We believe that a thorough study of this single, large-scale model during its training can already provide valuable insights. Our perspective is that understanding why a ResNet50 generalizes on ImageNet is already a significant challenge. By focusing on a single model, we aim to derive credible hypotheses in the study of simple and complex features, their order of appearance, types of features, and the link between complexity and generalization. However, we agree that this study should be extended to other architectures in future work to generalize these findings.

---

### Comment · Area_Chair_GioD · 2024-08-12

Dear reviewers,

The discussion period will end soon. Please read the rebuttal from the authors and participate in the discussion, as soon as possible.

Thank you.

Best,

AC

---

### Decision · Program_Chairs · 2024-09-25

**Decision:**

Accept (poster)

**Comment:**

This paper proposes a new feature complexity metric for deep neural networks, and explores the learning dynamics of features with different complexities, the flow of different features in forward propagation, and the importance of different features to the output. After an active discussion, three reviewers leaned towards acceptance, while two leaned towards borderline rejection. The reviewers praised the good presentation of the paper, the novelty of the proposed feature complexity metric, and the thoroughness of the related literature. The reviewers were concerned about the generalizability of the findings as the paper only conducted experiments on the ResNet-50 model, and the lack of comparison between the proposed metric with other feature complexity metrics. One reviewer was also concerned about significance of the findings, while another reviewer held an opposite view that the paper added significant depth to existing knowledge. During the rebuttal period, the authors discussed about the limitations of only studying the ResNet-50 model, and compared between the proposed metric and other metrics (e.g., the CKA metric). The reviewers feel most of their concerns are addressed after the rebuttal period.